# Orphan GPR116 mediates the insulin sensitizing effects of the hepatokine FNDC4 in adipose tissue

Anastasia Georgiadi [1,2,3,4,5✉], Valeria Lopez-Salazar[1,2,3,4], Rabih El-Merahbi [1,2,3,4],
Rhoda Anane Karikari[1,2,3,4], Xiaochuan Ma[5], André Mourão[6], Katarina Klepac[1,2,3,4], Lea Bühler[1,2,3,4],
Ana Jimena Alfaro[1,2,3,4], Isabell Kaczmarek[7], Adam Linford[1,2,3,4], Madeleen Bosma [5], Olga Shilkova [5],
Olli Ritvos[8], Nobuhiro Nakamura [9], Shigehisa Hirose[9], Maximilian Lassi[4,10], Raffaele Teperino [4,10],
Juliano Machado[1,2,3,4], Marcel Scheideler [1,2,3,4], Arne Dietrich[11], Arie Geerlof[6], Annette Feuchtinger[12],
Andreas Blutke[12], Katrin Fischer[13], Timo Dirk Müller [13], Katharina Kessler[4,14,15,16], Torsten Schöneberg [7],
Doreen Thor [7], Silke Hornemann[4,14], Michael Kruse[14,15], Peter Nawroth [1,2,4],
Olga Pivovarova-Ramich[4,14,15,17], Andreas Friedrich Hermann Pfeiffer[4,14,15], Michael Sattler [6],
Matthias Blüher[18] & Stephan Herzig [1,2,3,4✉]

The proper functional interaction between different tissues represents a key component in systemic metabolic control. Indeed, disruption of endocrine inter-tissue communication is a hallmark of severe metabolic dysfunction in obesity and diabetes. Here, we show that the FNDC4-GPR116, liver-white adipose tissue endocrine axis controls glucose homeostasis. We found that the liver primarily controlled the circulating levels of soluble FNDC4 (sFNDC4) and lowering of the hepatokine FNDC4 led to prediabetes in mice. Further, we identified the orphan adhesion GPCR GPR116 as a receptor of sFNDC4 in the white adipose tissue. Upon direct and high affinity binding of sFNDC4 to GPR116, sFNDC4 promoted insulin signaling and insulin-mediated glucose uptake in white adipocytes. Indeed, supplementation with FcsFNDC4 in prediabetic mice improved glucose tolerance and inflammatory markers in a white-adipocyte selective and GPR116-dependent manner. Of note, the sFNDC4-GPR116, liver-adipose tissue axis was dampened in (pre) diabetic human patients. Thus our findings will now allow for harnessing this endocrine circuit for alternative therapeutic strategies in obesity-related pre-diabetes.

A full list of author affiliations appears at the end of the paper.

Type 2 diabetes (T2D) is a gradually developing disease in which genetic, lifestyle, and ageing factors each may separately, or in combination, accelerate its progression and severity. Early glucose intolerance is a hallmark of the pre-diabetic state and it is targeted therapeutically by the prescription of metformin or lifestyle changes, such as diet and exercise[1,2]. However, there is a lack of mechanistic understanding of pre-diabetes development, as well as the endocrine and molecular factors, that can explain the individual responsiveness to therapeutic interventions.

Of the plethora of biological responses elicited by insulin, the removal of excess glucose from the blood circulation is central in the pathogenesis of obesity-related T2D. Progressive insulin resistance and the subsequent failure to cope with dietary glucose, i.e., glucose intolerance, mostly reflects the inability of the adipose tissue (AT) and skeletal muscle to sufficiently eliminate circulating glucose in response to insulin. In this pre-diabetic state, improvements in insulin sensitivity are in principal still able to delay or even prevent the onset of full-blown diabetes, making this disease stage an attractive target for tailored interventions and preventive measures. However, mechanisms contributing to the glucose-intolerant, pre-diabetic phenotype remain vastly unclear.

Recent research progress has clearly promoted the concept that systemic glucose homeostasis is determined by a variety of inter-tissue communication pathways, and most peripheral organs have been described to exhibit a secretory, endocrine function, including the liver. Indeed, hepatic steatosis is the strongest predictor of insulin resistance in skeletal muscle and AT, tightly coupled to alterations in the hepatic secretory function and the release of so-called hepatokines that are capable of controlling distant metabolic processes[3]. Understanding of specific endocrine routes by which the liver regulates insulin action in distinct peripheral organs has just emerged, with critical mechanisms remaining to be discovered.

Fibronectin type III domain containing 4 (FNDC4) is a type I transmembrane protein, which has been demonstrated to release a soluble bioactive protein that is highly conserved among mouse and primates[4]. Soluble FNDC4 (sFNDC4) has been reported to exert anti-inflammatory effects on macrophages[4], osteoclasts[5], and adipocytes[6], promoting survival in response to severe chronic inflammation[4] and improving insulin resistance[6]. Our understanding of FNDC4 biology has only just begun and it is unclear to date whether this secreted protein acts as a hormone. In this regard, the receptor and its signaling, as well as target organs of sFNDC4 remain unknown; it is unclear whether and how sFNDC4 acts in an endocrine manner to maintain glucose homeostasis. Here, we report an as-yet undiscovered physiological role of FNDC4 as a hepatokine. We show that the liver primarily controlled the circulating levels of sFNDC4 and that lowering of hepatic FNDC4 resulted in a pre-diabetic phenotype in mice. Indeed, we demonstrate that circulating levels of sFNDC4 in several human and mouse cohorts showed tight correlation with insulin sensitivity. In addition, we now have identified the orphan adhesion G protein-coupled receptor 116 (GPR116) as a receptor of sFNDC4 in white adipose tissue (WAT), thereby establishing an endocrine FNDC4–GPR116 axis in the control of systemic glucose homeostasis. Intriguingly, this axis was impaired in diabetic patients and therapeutic injections of recombinant FcsFNDC4 into pre-diabetic mice corrected pre-diabetic hyperglycemia, now providing a rationale for harnessing the FNDC4–GPR116 axis in pre-diabetes therapy.

## Results

**Liver and serum FNDC4 levels positively associate with glucose tolerance in humans.** Using tissue mRNA profiling, we found *Fndc4* mRNA to be most highly expressed in the liver and brain

of mice and humans (Fig. 1a). To investigate the association of FNDC4 with glycemic control in humans, we measured the mRNA levels of liver and AT *FNDC4* from lean and obese humans with or without T2D (see "Methods: Cross-sectional study—Leipzig"). Liver *FNDC4* mRNA levels showed an inverse correlation with fasting blood glucose levels (Fig. 1b and Supplementary Table 1) and blood glucose levels after a 2-h oral glucose tolerance test (OGTT) (Fig. 1c and Supplementary Table 1) in lean healthy individuals. Since lower levels of blood glucose in those tests indicate increased glucose tolerance, these findings proposed a positive correlation between *FNDC4* levels and glucose tolerance, as well as insulin sensitivity. In addition, liver *FNDC4* mRNA levels decreased in obese humans with impaired glucose tolerance and impaired insulin tolerance (IGT/IIT) and in obese subjects with clinically diagnosed T2D compared to normoglycemic, non-diabetic (ND) lean controls (Fig. 1d and Supplementary Table 1).

FNDC4 has been shown to release a soluble peptide (sFNDC4)[4] and there are limited reports of sFNDC4 levels in the blood circulation of humans or mice. Therefore, we quantified sFNDC4 in human serum in a cohort of healthy individuals receiving a high-fat diet (HFD), isocaloric to the control diet, for 6 weeks. Blood was collected for analysis after 1 and 6 weeks (see "Methods: NUGAT study"). Under this diet, the participants did not gain weight, but showed an increased Homeostasis Model Assessment (HOMA) index, indicating the appearance of insulin resistance in response to the consumed HFD[7]. We observed a 10% decrease and a sustained decrease of 6% compared to baseline (LF), in the serum sFNDC4 levels, following 1 week and 6 weeks HFD, respectively (Fig. 1e). Thus, under these early pre-diabetes settings, circulating levels of sFNDC4 showed a significant decrease in humans. Overall, these data supported a positive correlation between liver *FNDC4* mRNA, serum sFNDC4 levels, and glucose tolerance as well as insulin sensitivity in humans (See also Supplementary Note 1: FNDC4 ELISA-FNDC4 quantification in plasma or serum- ELISA Validation and related Supplementary Fig. 1.).

**Hepatic FNDC4 is required for proper systemic glucose tolerance and specifically targets WAT.** To determine the role of hepatic FNDC4 in glucose homeostasis, we lowered hepatic FNDC4 levels using an AAV8-shFNDC4 specifically targeting the liver. To confirm the knockdown (KD) effect of the adeno-associated virus (AAV), we measured liver and circulating FNDC4 3 weeks post AAV injection and then split the AAVsh-Control and AAVshFNDC4-injected animals into HFD or chow diet groups for a total of 8 weeks (Fig. 2a). Three weeks after the delivery of AAVshFNDC4, liver *Fndc4* mRNA decreased by 40% (Fig. 2b), and both liver FNDC4 protein (Fig. 2c) and sFNDC4 plasma levels (Fig. 2d) were significantly reduced. In addition, *Fndc4* mRNA was not altered in non-hepatic tissues, such as gonadal WAT (gWAT) and skeletal muscle (gastrocnemius muscle) (Fig. 2b), supporting the notion that the liver represents the main source of circulating FNDC4. Finally, at the end of the study (8 weeks on HFD and total 11 weeks post AAV injections), liver *Fndc4* mRNA (Fig. 2e) as well as circulating levels of sFNDC4 (Fig. 2f) still remained significantly reduced in the AAVshFNDC4 group compared to the AAVshControl group under chow and HFD conditions. Of note, we observed substantial differences with regard to the quantified levels of sFNDC4 in mouse plasma derived from trunk blood as opposed to tail blood, with tail plasma measurements of sFNDC4 being up to 10 times lower compared to trunk-derived plasma (see Supplementary Note 1 and Supplementary Figs. 1g and 2a, b).

Under chow diet, AAVshFNDC4 treated mice showed no difference in glucose clearance during an intraperitoneal glucose

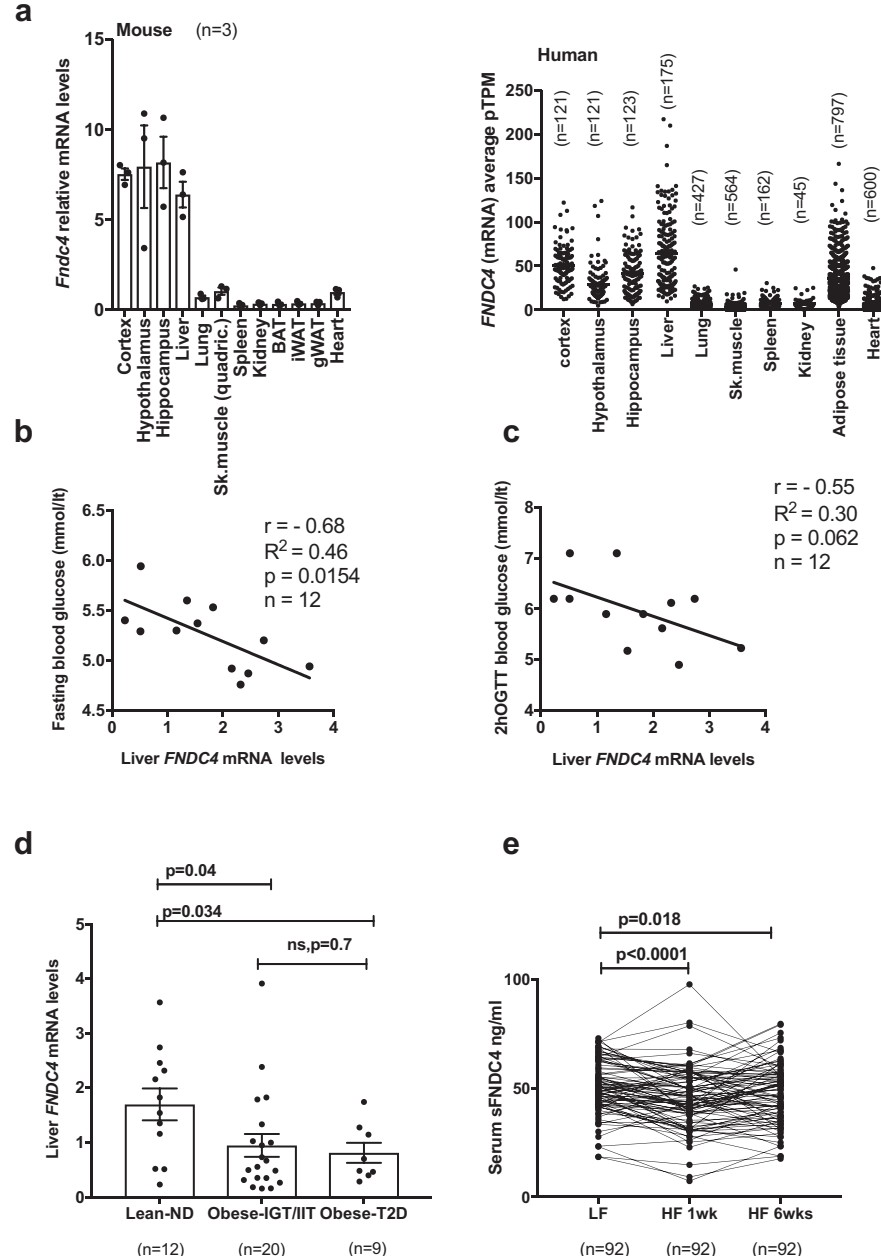

**Fig. 1 Liver and serum FNDC4 levels positively associate with glucose tolerance in humans. a** RT-qPCR quantification of mRNA levels of the mouse (left panel, $n = 3$ mice per tissue, male, C57Bl6N, 7–8 weeks old, skeletal muscle is quadriceps) and the human *Fndc4* gene in the indicated tissues (right panel, *n* are shown on the figure and represent independent humans). Left panel: 2^ddCt values are shown, which are expressed relative to *Fndc4* mRNA levels for the Sk.muscle group. *Tbp* is used as a housekeeping gene. Right panel: The human data were retrieved from the Protein Atlas Project database (URL: http://www.proteinatlas.org/search/Fndc4) and represent pTMP (read counts normalized to transcripts per million coding genes). Exact values are provided in the Source Data. Pearson's correlation for human liver *FNDC4* mRNA levels and fasting blood glucose levels (mmol/l) (**b**) and **c** blood glucose levels (mmol/l) 2 h post oral glucose tolerance test (OGTT) ($n = 12$ independent humans). **d** RT-qPCR quantification of human liver *FNDC4* mRNA levels at the indicated groups. ND non-diabetic, T2D type 2 diabetes, Obese-IGT/IIT impaired glucose tolerance/impaired insulin tolerance ("Methods: Cross-sectional study—Leipzig") (*n* represents independent humans and exact numbers per group are shown on the figure). Statistics represent unpaired two-tailed *t* test. mRNA levels are provided as mRNA quantity (calculated based on the standard curve method). **e** Serum levels of sFNDC4 ng/ml in paired blood samples from humans, who initially consumed a low-fat (LF) diet for 6 weeks and subsequently were given a high fat (HF) diet for 6 weeks ("Methods: NUGAT study-DIfE"). Serum was collected at the end of the LF diet period (LF) and at 1 week (HF 1 week) and 6 weeks (HF 6 weeks) of HFD diet (paired samples), $n = 92$ independent humans. Statistics represent paired, two-tailed *t* test. Data shown are mean ± SEM.p *p*-value, ns non-significant. Source data are provided as a Source data file.

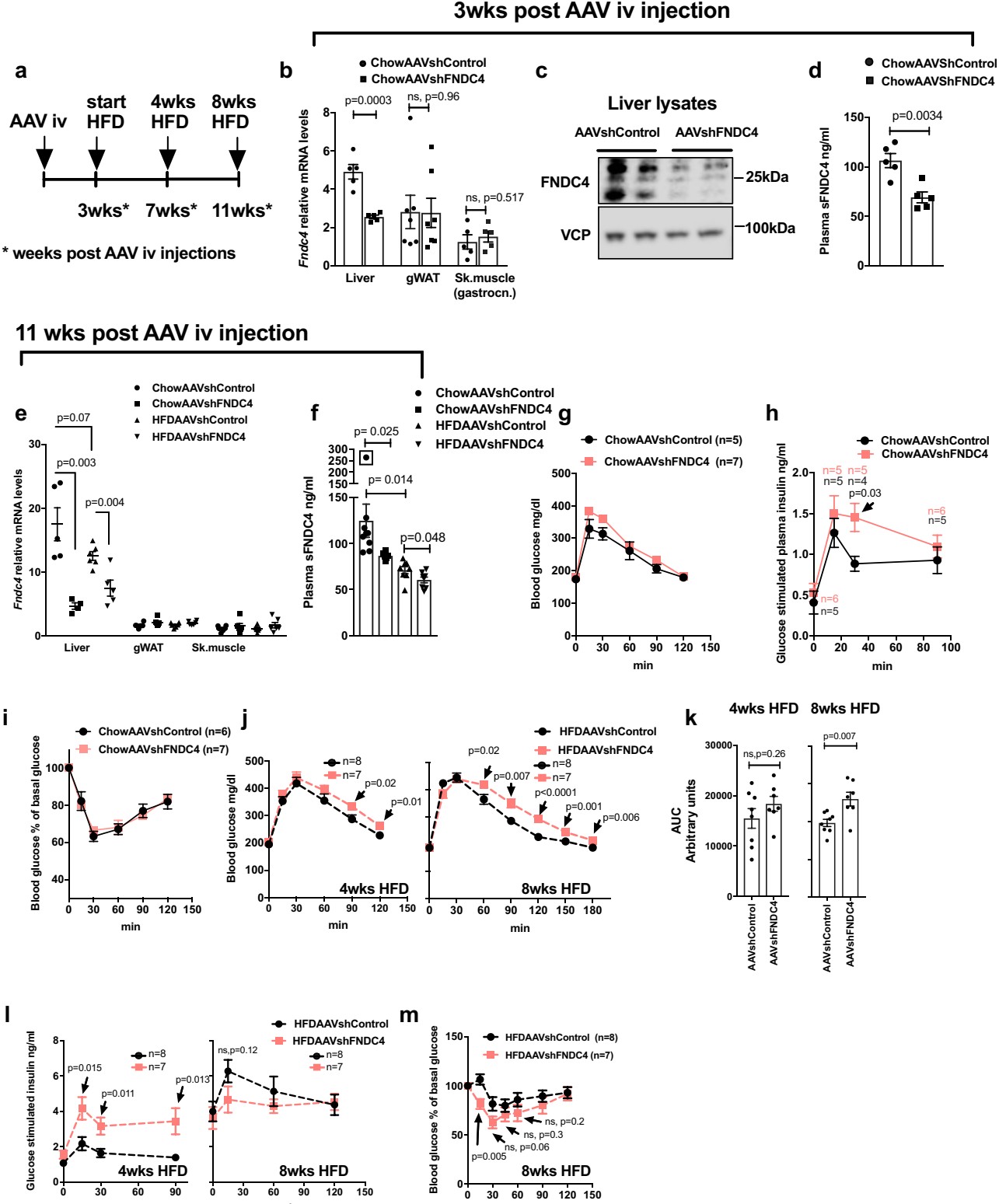

tolerance test (IPGTT) (Fig. 2g); however, they exhibited compensatory hyperinsulinemia during the IPGTT (Fig. 2h) and showed no significant difference during an insulin tolerance test (ITT) compared to the AAVshControl mice (Fig. 2i). Furthermore, AAVshFNDC4 mice on HFD tended to have higher blood glucose at 4 weeks of HFD and showed impaired glucose clearance at 8 weeks of HFD compared to the AAVshControl animals, during the IPGTT (Fig. 2j, k).

Importantly, at 4 weeks on HFD AAVshFNDC4 showed severe compensatory hyperinsulinemia during the IPGTT (Fig. 2l). There was no significant difference between the two groups in the ITT at 8 weeks on HFD (Fig. 2m).

To examine the possible effects of AAVshFNDC4 liver deletion on insulin secretion, we also examined the levels of fasting and glucose-stimulated C-peptide and found no difference between AAVshFNDC4 and AAVshControl groups, under chow or HFD

**Fig. 2 Mice with hepatic deletion of FNDC4 exhibited decreased circulating levels of sFNDC4 and developed a pre-diabetes phenotype. a** Schematic representation of the study protocol. **b** RT-qPCR quantification of *Fndc4* mRNA at the indicated tissues ($n = 5$ mice per group for the liver and skeletal muscle (gastrocnemius) and $n = 7$ mice per group for gonadal white adipose tissue (gWAT). **c** Western blot against FNDC4 protein in the liver and VCP as loading control ($n = 2$ mice per group are shown). Results on additional mice and uncropped blots are shown in Source data. **d** ELISA quantification of plasma (trunk blood) sFNDC4 ($n = 5$ mice per group), 3 weeks post intravenous (iv) injection of AAVs. **e** RT-qPCR quantification of *Fndc4* mRNA at the indicated tissues ($n = 5$ mice for Liver ChowAAVshControl, Liver HFDAAVShFNDC4, $n = 4$ mice for Liver ChowAAVshFNDC4, and $n = 6$ mice for Liver HFDAAVshControl, gWAT ChowAAVshControl, gWATChowAAVshFNDC4, gWAT HFDAAVshControl, gWAT HFDAAVshFNDC4, Sk.muscle ChowAAVshControl, Sk.muscle ChowAAVshFNDC4, Sk.muscle HFDAAVshControl, Sk.muscle HFDAAVshFNDC4). Skeletal muscle is gastrocnemius. For **b**, **e**, 2^ddCt values are shown, which are expressed relative to the Sk.muscle group. *Hprt* is used as a housekeeping gene for liver and gWAT and *H3f3* is used as a housekeeping gene for Sk.muscle. **f** ELISA quantification of plasma (trunk blood) sFNDC4 ($n = 8$ mice ChowAAVshControl, $n = 6$ mice ChowAAVshFNDC4, $n = 7$ mice HFDAAVshControl, $n = 8$ mice HFDAAVshFNDC4) 11 weeks post iv injection of AAVs and 8 weeks on HFD or chow control diet. Statistics represent unpaired two-tailed *t* test. Data point in square was not taken into account for the statistics, due to very high levels of sFNDC4 compared to the rest of the mice in the group (outlier). Glucose homeostasis in chow and HFD fed mice: **g** blood glucose and **h** plasma insulin levels during an IPGTT. **i** blood glucose levels during an ITT on chow-fed mice. **j** Blood glucose levels, **k** area under the curve (refers to **j**), and **l** plasma insulin during an IPGTT on HFD-fed mice (weeks on HFD as shown on graph). **m** Blood glucose levels during an ITT on HFD-fed mice. In **g**, **h**, **j**, **m**, $n =$ independent mice and the exact number of mice is shown on the figure. Male, C57BL6N mice were put on HFD when 10–11 weeks old and compared with age-matched chow fed mice. HFD contained 45% fat. During IPGTT and glucose-induced insulin test, 2 g/kg D-glucose was injected (ip) and during ITT 0.8 U/kg insulin was used (ip). p p-value, ns non-significant. Data shown are mean ± SEM. Statistics in **b**, **d–f**, **h**, **j–m** represent an unpaired, two-tailed Student's *t* test. Source data are provided as a Source data file.

conditions (Supplementary Fig. 2c, d). Furthermore, the insulin content in the pancreas of AAVshFNDC4 mice was not significantly different from that of the AAVshControl group on chow or HFD (Supplementary Fig. 1e), nor did recombinant FNDC4 alter insulin secretion in primary pancreatic islets in vitro (Supplementary Fig. 2f). Our findings suggested an intact pancreatic function in the AAVshFNDC4 treated group and argued for an increased peripheral insulin resistance resulting in hyperinsulinemia compared to the AAVshControl treated group. Finally, no differences in body weight (Supplementary Fig. 2g, j), organ weights (Supplementary Fig. 2i, l), and food intake (Supplementary Fig. 2h, k) between AAVshFNDC4 compared to AAVshControl were observed neither under chow nor under HFD conditions. Furthermore, there were no significant differences in liver or muscle triglyceride (TG) content between AAVshFNDC4 and AAVshControl under chow or HFD diet (Supplementary Fig. 2m, n). Serum cholesterol (Supplementary Fig. 2o), TG (Supplementary Fig. 2p), and non-esterified fatty acid (NEFA) levels (Supplementary Fig. 2q) also remained unchanged between AAVshFNDC4 and AAVshControl groups, under HFD conditions. Overall, these findings demonstrate that decreasing liver and circulating FNDC4 promoted a state of pre-diabetes, manifested by glucose intolerance combined with compensatory hyperinsulinemia and subsequent hyperglycemia.

To identify the tissue target(s) of sFNDC4, we injected recombinant mammalian, long-lived FcsFNDC4 to HFD mice with glucose intolerance and traced tissue glucose uptake after long-term injections. To determine the injected dose of recombinant sFNDC4, we examined the circulating levels of sFNDC4 in mice under chow and HFD feeding. sFNDC4 was present in the circulation throughout the day but tended to peak several hours before the mice entered the feeding/dark phase (Fig. 3a). Remarkably, HFD feeding reduced the circulating levels of sFNDC4 (Fig. 3a) and decreased liver mRNA levels of *Fndc4* (Fig. 3b). We found that intraperitoneal (ip) injections of long-lived FcsFNDC4, at a dose of 0.2 mg/kg every second day, recovered the decreased levels of sFNDC4 in the HFD group to the normal levels of the chow fed group (Fig. 3c). Therefore, we used the dose of 0.2 mg/kg, injected every second day to treat HFD-fed mice. Under these conditions, we observed an improvement in glucose tolerance after 2 weeks (Fig. 3d, e), which was maintained for up to 4 weeks upon injections (Fig. 3d, e). We saw no differences in glucose-stimulated insulin secretion (Fig. 3f) and insulin tolerance (ITT) (Fig. 3g) between FcsFNDC4

and vehicle control (VC)-treated mice. Also, body weight (Supplementary Fig. 3a) and food intake (Supplementary Fig. 3b) were not altered. Furthermore, we found no difference in organ weights (Supplementary Fig. 3c) or in liver and muscle TGs (Supplementary Fig. 3d–f), which suggested that changes in lipid content in those metabolic tissues could not have accounted for the improved glucose clearance during the IPGTT upon chronic FcsFNDC4 injections.

To examine the tissues contributing to the improved glucose clearance during the IPGTT, we evaluated the glucose uptake in different tissues using fluorescently labeled glucose (2-NBDG (2-(N-(7-nitrobenz-2-oxa-1,3-diazol-4-yl)amino)-2-deoxyglucose)). We found a significantly higher uptake of fluorescent glucose in the gWAT of HFD mice injected with FcsFNDC4 compared to VC (Fig. 3h). In addition, we only observed an increase in WAT pAKT levels of FcsFNDC4-treated HFD mice (4 weeks) compared to VC after a single ip injection of insulin, whereas this effect was absent in liver and skeletal muscle (Fig. 3i), supporting an insulin-sensitizing effect of FcsFNDC4 specifically in WAT.

Closer histological examination of the gWAT showed no difference in adipocyte size (Fig. 3j) but reduced Cd68-positive cells upon FcsFNDC4 injections (Fig. 3k, l), suggesting a reduction of macrophages in gWAT of FcsFNDC4-injected mice compared to VC controls. In addition, FscFNDC4 mice had lower total tissue *Cd68* mRNA in the gWAT compared to VC controls (Fig. 3m). mRNA levels of *Resistin*, *Tnfalpha* (tumor necrosis factor-α), *Mcp-1* (monocyte chemoattractant protein-1), *Ccl11* (CC motif chemokine 11), *Il10* (interleukin 10), *Il6* (interleukin-6), and *Cd206* (cluster of differentiation 206) were decreased in gWAT of FcsFNDC4 versus VC treated mice (Fig. 3n), indicating a reduced inflammatory status in this tissue upon FcFNDC4 delivery. Furthermore, FcsFNDC4-treated mice exhibited reduced levels of circulating TNFalpha (Fig. 3o) and Resistin proteins (Fig. 3p), the latter being specifically secreted from WAT in mice[8]. We found no difference in circulating leptin (Supplementary Fig. 3g) and adiponectin (Supplementary Fig. 3h) between FcsFNDC4-injected mice compared to VC group. To investigate whether those effects were associated with improvement of endoplasmic reticulum (ER) stress as it has been suggested for sFNDC4 before[6], we quantified ER stress markers, such as *Edem* (ER-degradation-enhancing-α-mannidose-like protein), *Xbp1* (X-box binding protein 1), *Grp78* (Glucose regulated protein 78), *Atf6* (activating transcription factor 6), *Grp94* (Glucose regulated protein 94), *DNAjb9* (DnaJ Heat Shock Protein Family Member 9), *Erdj4*

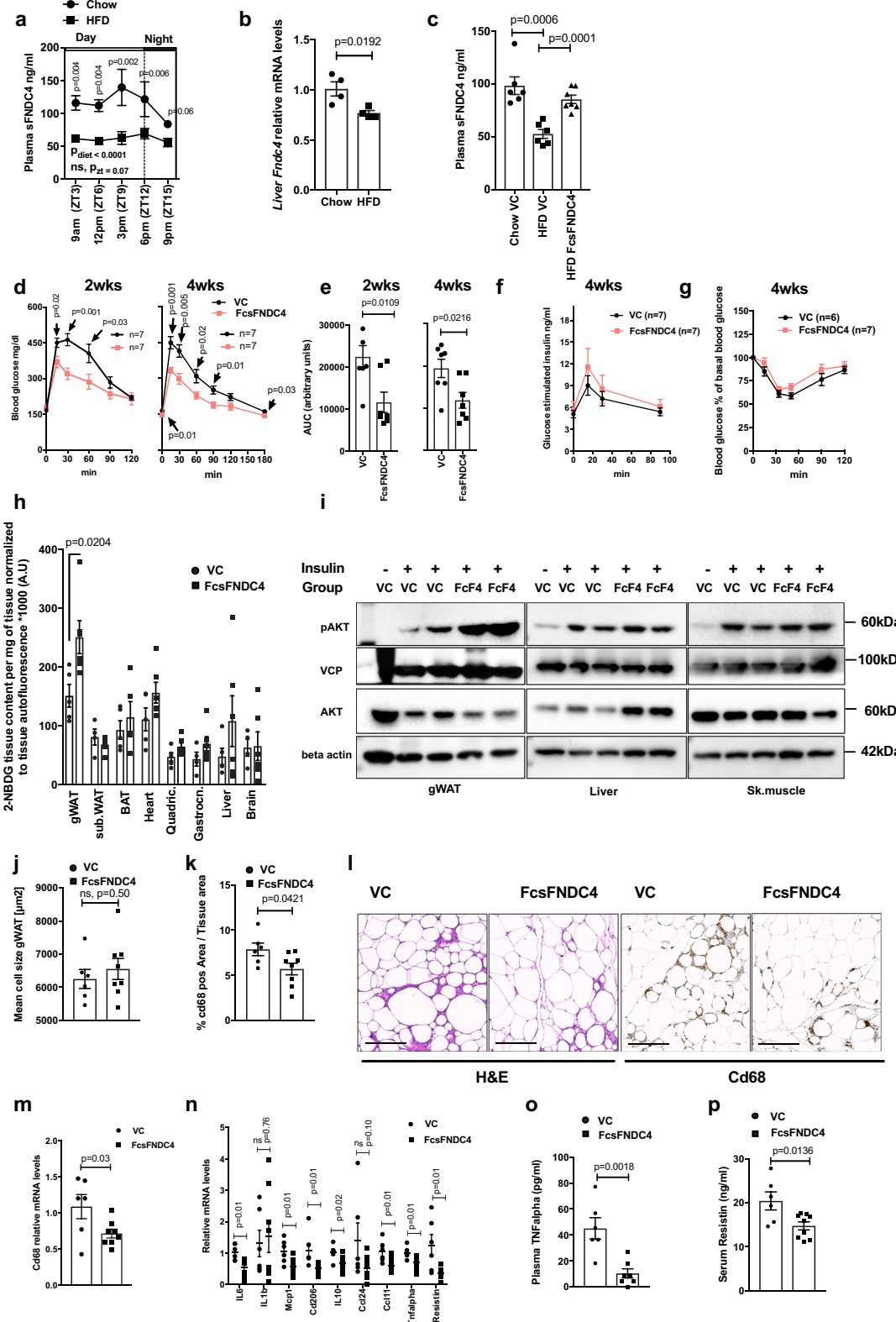

(ER-localized DnaJ type II homolog 4), *Chop* (CCATT-enhancer-binding protein homologous protein), as well as a regulator of ER stress, *Hmox1* or *HO-1* (heme oxygenase 1), in the WAT tissue of FcsFNDC4- and VC-treated HFD mice, but we observed no difference in the expression levels of ER stress gene markers between those groups (Supplementary Fig. 3i). In addition, to assess FcsFNDC4 on ER stress directly on adipocytes, we incubated

3T3L1 adipocytes with palmitate (PA) (200 μM) and FcsFNDC4 (200 nM) or Fc control (200 nM) for 24 h and blotted against activated master ER stress regulator inositol-requiring enzyme 1 (IRE1). We found no impact of FcsFNDC4 on the protein levels of PA-induced pIRE1 on 3T3L1 adipocytes treated 24 h with PA, as opposed to Fc control plus PA treated group (Supplementary Fig. 3j). Thus, our findings do not suggest that the amelioration of

**Fig. 3 Every second day, injections of rec. FcsFNDC4 0.2 mg/kg improved glucose tolerance and increased glucose uptake specifically in the white adipose tissue. a** Plasma (trunk) sFNDC4 in mice on 16 weeks HFD ($n = 4$ independent mice per time point) and Chow control mice ($n = 2-4$ independent mice per time point). Statistics represent a regular two-way ANOVA for diet (Pdiet) and time (Pzt). Multiple comparisons for diet effect were performed according to Holm–Sidak's test and exact p values are shown for chow–HFD diet comparison for each time point. **b** RT-qPCR quantification of liver *Fndc4* mRNA at 16 weeks HFD and Chow mice ($n = 4$ mice per group). **c** Plasma (trunk) sFNDC4 ng/ml from chow and 14 weeks HFD, 48 h after a single ip injection of FcsFNDC4 (0.2 mg/kg) or VC ($n = 6$ mice per group for Chow VS and HFD VC, $n = 7$ mice per geup for HFD FcsFNDC4). **d** Blood glucose during IPGTT test, **e** area under the curve (AUC), **f** glucose-stimulated insulin response (during the IPGTT) in **d**, **e** (**d-f** $n = 7$ mice per group), and **g** percentage of blood glucose levels during an ITT using 0.8 U/kg insulin ip, (VC—$n = 6$ mice, FcsFNDC4—$n = 7$ mice) on HFD, 2 and 4 weeks injected with VC or FcsFNDC4. **h** Quantification of 2-NBDG glucose content at shown tissues, 35 min after ip 2-NBDG injection ($n = 4-6$ mice per group, exact numbers per group are provided in Source data). **i** Western blot of pAKT(Ser473) and total AKT at the indicated tissues, after an acute injection of insulin. pAKT and AKT were run on different blots, with VCP and beta actin as loading controls, respectively. This experiment was performed once under the exact same conditions. **j** Mean cell size of adipocytes and **k** percentage of Cd68-positive area of gWAT (**j**, **k** VC—$n = 6$ mice, FcsFNDC4—$n = 8$ mice). **l** Representative images (out of 20 images per mouse, VC—$n = 6$ mice, FcsFNDC4—$n = 8$ mice) for hematoxylin and eosin (H&E) and anti-Cd68 staining of gWAT. Scale bar = 200 μm. **m**, **n** RT-qPCR quantification of mRNA levels of the presented genes in gWAT (VC—$n = 6$ mice and FcsFNDC4—$n = 8$ mice). Data shown represent 2^ddCt values. H3f3 was used as a housekeeping gene. ELISA quantification of **o** plasma TNFalpha (VC—$n = 6$ mice and FcsFNDC4-—$n = 7$ mice) and **p** serum resistin (VC—$n = 6$ mice and FcsFNDC4—$n = 9$ mice). For **h–p** HFD mice, 4 weeks injected with VC or FcsFNDC4 are shown. HFD contained 60% fat and it was initiated at 7–8 weeks of age. Mice were males, C57BL6N. Data shown are mean ± SEM. Statistics are unpaired, two-sided *t* test. p p-value, ns non-significant. Source data are provided as a Source data file.

WAT insulin signaling by FcsFNDC4 derived from alleviation of ER stress as previously published by Lee et al.[6].

Finally, to exclude any effects of FcFNDC4 on energy expenditure (EE), also linked to alterations in systemic inflammation, we measured EE in either HFD-fed mice treated chronically with FcFNDC4 (Supplementary Fig. 3k–q) or in lean young, chow diet fed mice 2 h after a single injection of FcsFNDC4 (Supplementary Fig. 3r, s). In both cases, no differences in EE per se or in correlation to body weight between FcsFNDC4-injected and control mice was observed (Supplementary Fig. 3k–m and Supplementary Fig. 3r), as there were also no differences in locomotor activity (Supplementary Fig. 3q, s), respiratory exchange ratio (RER; Supplementary Fig. 3p), and oxygen consumption (Supplementary Fig. 3o). Also, we observed no differences in cumulative food consumption (data not shown). Thus, our findings supported a specific action of FcsFNDC4 in the WAT, independent of EE but linked to improved glucose uptake, improved insulin signaling, and reduced levels of pro-inflammatory adipokines, as well as a local reduction of macrophages in the gWAT.

**GPR116 acts as a receptor for sFNDC4.** To initially identify candidate receptors for sFNDC4, we set up a fluorescence-readout binding assay in live cells. To this end, we utilized recombinant sFNDC4, corresponding to the extracellular part of FNDC4 protein (mouse FNDC4 aa: 40–160) fused with human IgG (Fc). The binding of FcsFNDC4 to cells was quantified by detecting the cell-bound ligand (FcsFNDC4) with secondary IgG-PE (phycoerythrin) conjugated antibody using fluorescent flow cytometry. For screening, we chose immortalized mouse pre-adipocytes (imm. SVF) from inguinal WAT[9] due to their higher survival and robustness during the fluorescence-activated cell sorting (FACS) staining/sorting protocol, the possibility to sort higher numbers, and to ensure reproducibility. We performed saturation binding with increasing concentrations of FcsFNDC4 from 0 to 500 nM (see also Supplementary Note 2). We assessed the total population of cells (Supplementary Fig. 4a) and we observed increasing levels of fluorescence intensity (phycoerythrin (PE)) following the increasing FcsFNDC4 concentration (Supplementary Fig. 4b). By performing a saturation binding curve, we observed saturation of fluorescence readouts around 100 nM of FcsFNDC4. In contrast to FcsFNDC4, the binding of Fc control did not show any saturation of fluorescence (Fig. 4a). We thus used Fc as a negative control in our assays. Furthermore, by utilizing a competitive sFNDC4-binding assay we observed

a sFNDC4 concentration-dependent reduction on the binding of FcsFNDC4, supporting the idea of specific binding to a membrane receptor (Fig. 4b).

Next, high- and low-binding cell populations (HBC and LBC, respectively; Fig. 4c) were sorted, expanded, and re-sorted for up to 20 passages to obtain cell populations with stable high and low binding properties to FcsFNDC4 (Fig. 4d). These final HBC and LBC were analyzed for differentially expressed genes by Affymetrix transcriptomics analysis. Among the top 30 upregulated genes, we identified two G protein-coupled receptors (GPCRs) and an integrin receptor to be more highly expressed in HBC compared to LBC: relaxin/insulin-like family peptide receptor 1 (*Rxfp1*, fold change = 16), G protein-coupled receptor 116 (*Gpr116*, fold change = 10), and integrin subunit alpha D (*ItgaD*, fold change = 9) (Fig. 4e). RXFP1 ligands are relaxins and insulin-like peptide 3 (INSL3)[10]. GPR116 is an orphan adhesion GPCR and integrin receptors are known to interact with FN3 domain-containing proteins[11,12]. To test the impact of these candidate receptors on FcsFNDC4 binding, we transiently overexpressed human *RXFP1, GPR116* or *ITGAD* in HEK239A cells (Fig. 4g), a cell line that showed much lower baseline binding to FcsFNDC4 than the immortalized pre-adipocytes (Supplementary Fig. 4c, d). *GPR116* overexpression (OE) increased the binding of FcsFNDC4 compared to control, whereas *RXFP1* or *ITGAD* OE did not have any impact on FcsFNDC4 binding (Fig. 4f). Ligand binding to integrin receptors requires divalent cations such as $Ca^{2+}$ and $Mg^{2+}$[13]. Thus, to further test whether an integrin receptor could possibly mediate the binding of FcFNDC4, we performed FcsFNDC4 binding in the presence of increasing concentrations of EDTA chelator, thereby inhibiting ligand binding to integrins. We did not observe any effect of EDTA on FcsFNDC4 binding. Only very high concentrations of EDTA (10 mM) abolished binding (Fig. 4h), overall supporting the hypothesis that GPR116 represents a specific sFNDC4 target receptor. Also, we confirmed the higher expression of GPR116 in HBC versus LBC at the mRNA and protein levels (Supplementary Fig. 4e).

To further assess the specificity of FcsFNDC4 binding to GPR116, we performed dose binding of FcsFNDC4 and Fc control to wild-type (WT), GRP116+/− (HET), and GPR116−/− (knock-out (KO)) SVF-derived mouse primary preadipocytes from the inguinal WAT fat depot. We found that FcsFNDC4 binding decreased upon reduced levels of *Gpr116* mRNA (Fig. 4i) suggesting that expression levels of *Gpr116* determined the binding of FcsFNDC4. To further assess direct and specific binding of

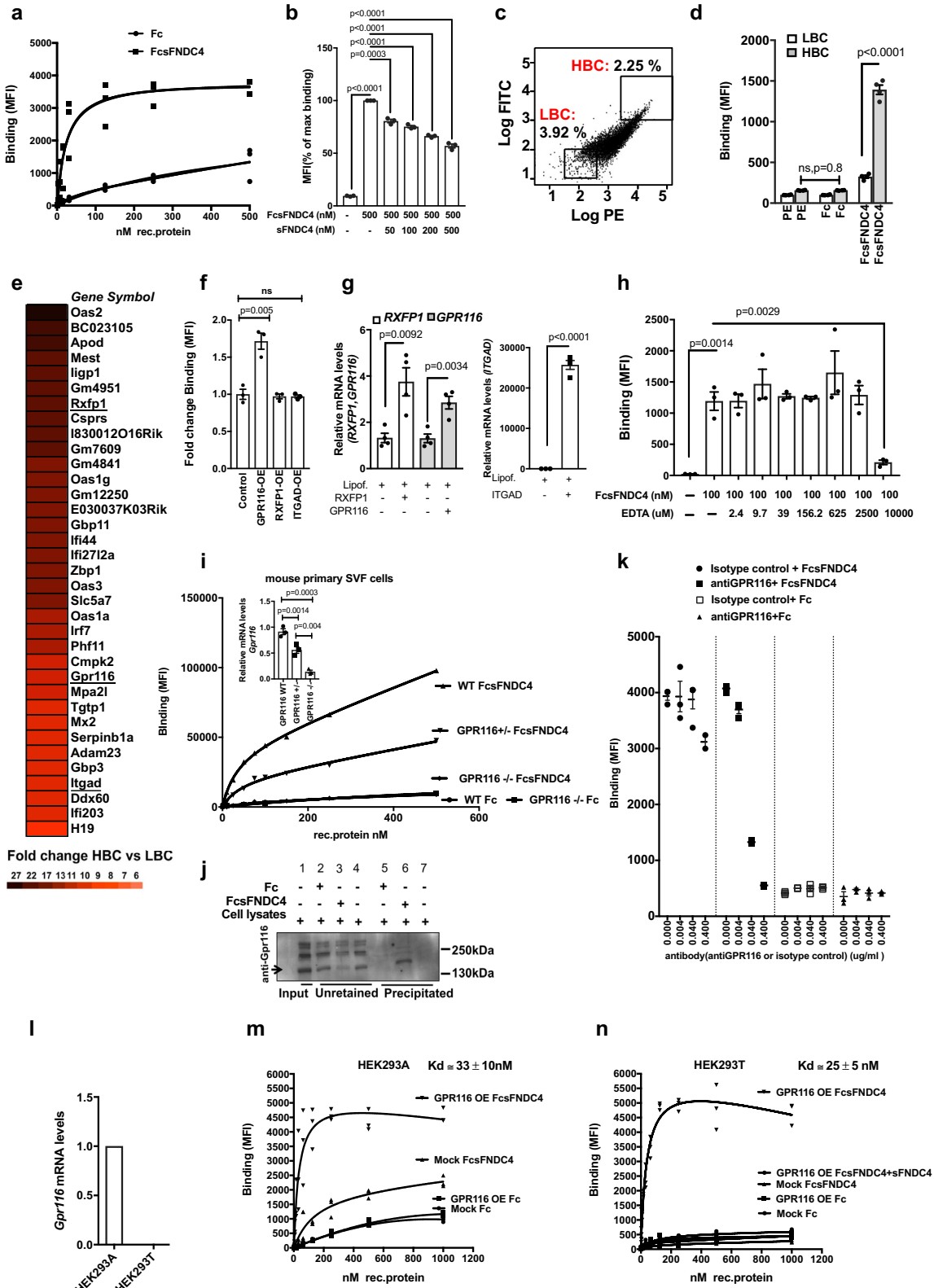

sFNDC4 to GPR116, we performed a GPR116 pull-down assay using FcsFNDC4 as bait. FcsFNDC4 precipitated GPR116 from total cell lysates of NIH3T3 cells, whereas there was no GPR116 precipitation with the Fc control (Fig. 4j). We have validated the specificity of the antiGPR116 (ab136262) used to detect GPR116 in Fig. 4j, in NIH3T3 preadipocytes with lenti-ShGPR116 KD and lenti-shControl. At cells with 70% deletion of GPR116 (shGPR116)

compared to control cells (shControl) (Supplementary Fig. 4f), this antibody showed no band close to 250 kDa and a much weaker band a bit higher than 130 kDa (all bands corresponding to the N-terminus of GPR116; Supplementary Fig. 4g).

FNDC4 and GPR116 crystal structures do not exist yet. Based on the interaction model of the GPR56 extracellular (EC) GAIN domain and Fibronectin type III monobody (FN3), which

**Fig. 4 Identification of GPR116 as a candidate receptor for sFNDC4. a** Saturation binding curves of FcsFNDC4 and Fc control to immortalized SVF (imm. SVF) preadipocytes derived from mouse iWAT. Y-axis shows ligand binding as mean fluorescence intensity (MFI) per 10,000 cells. Three replicate wells are shown per concentration of rec. protein. This experiment was repeated at least three times. **b** Competition of FcsFNDC4 binding (500 nM) with increasing concentrations of untagged sFNDC4. Data are expressed as percentage of max binding of 500 nM FcsFNDC4 ($n = 3$ independent experiments). Bars represent mean of three independent experiments ±SEM. **c** Gating of sorted cell populations of imm. SVF iWAT to very high log.PE/log.FITC signal (top 2.25% from total cell population = HBC-high binding cells) versus very low log.PE/log.FITC signal (bottom 3.92% from total cell population = LBC-low binding cells), positive for FcsFNDC4 binding (100 nM). This sorting was performed 20 times. **d** Binding of FcsFNDC4 (20 nM) and Fc control (20 nM) or only IgG-PE (PE) secondary in cells sorted from **c** after resorting and reseeding for 20 passages. Bars correspond to binding shown as MIF per 10,000 cells. This experiment was performed several times (20 times) up to passage 20 ($n = 4$ replicate wells). **e** Fold change mRNA expression data presented as a heat map. Top 35 genes identified with fold change HBC versus LBC >1.5 and $p < 0.05$ from Affymetrix gene expression arrays comparing LBC and HBC. Receptor genes are underlined. Color scale of fold change values is shown below the heat map. The means of three replicate wells per group (LBC and HBC) are compared. Absolute values and fold change calculations are presented in the Source data. The Affymetrix microarray data have been submitted to GEO (gene expression omnibus) under the identification GSE165329. Affymetrix arrays were performed once. **f** Fold change of binding (MFI) per 10,000 cells of FcsFNDC4 (100 nM) to HEK293A cells with transient overexpression of human RXFP1 or human GPR116 or human ITGAD. Binding to a mock transfection control (lipofectamine only) was used as control. $n = 3$ replicate wells of a representative experiment out of two independent experiments is shown. **g** (left and right panel) RT-qPCR quantification of the indicated genes at the indicated conditions. $n = 4$ replicate wells per condition are shown. This analysis was performed only once. TBP is used as a housekeeping gene. 2^ddCt values are shown. **h** Competition of FcsFNDC4 (100 nM) binding with increasing concentrations of EDTA ($0$–$10^3$ μM). Values are binding shown as MFI per 10,000 cells. $n = 3$ replicate wells of a representative experiment out of two independent experiments is shown. Cells are primary mouse SVF preadipocytes (iWAT). **i** Binding (MFI) of the indicated concentrations of FcsFNDC4 or Fc control to mouse SVF preadipocytes (iWAT) from WT, GPR116+/−, and GPR116−/− mice. $n = 1$ replicate well of cells pooled from $n = 7$–8 mice per genotype per tested concentration of rec. protein. This experiment was performed once. Mice were male, 7–9 weeks old. (**i**—top panel) RT-qPCR quantification of mRNA levels of Gpr116 in SVF iWAT preadipocytes used for binding in **i** at the indicated genotypes ($n = 3$ independent mice per genotype). Tbp is used as a housekeeping gene. **j** Pull down for GPR116: western blot analysis against GPR116. For pull down, 6xHis Dynabeads were prebound with 6xHis-Fc and 6xHis-Fc-sFNDC4 (30 μg), followed by incubation of 300 μg NIH3T3 cell lysates. (+) added to beads, (−) not added to beads, Input: rec. protein or cell lysates added to the Dynabeads. Unretained: cell lysates proteins that did not immunoprecipitate to the beads. Eluted: elution of GPR116, which precipitated on 6xHis-Fc-fused protein prebound to beads. Similar results were obtained from three independent experiments. Antibody against GPR116: ab136262. Representative WB out of three independent experiments is shown. Uncropped blot image is shown in Source data. **k** Mean fluorescence intensity (MFI) representing binding of FcsFNDC4 (100 nM) or Fc control (100 nM) to HEK293T GPR116 (human) OE cells in the presence of the indicated dose of antiGPR116 antibody, against the N-terminus of GPR116 (ab111169) or isotype control (ab171870). This experiment was performed once with $n = 3$ technical replicates. **l** q-PCR quantification of human Gpr116 mRNA levels in HEK293A (Ct = 25) and HEK293T (Ct = undetected) cells, $n = 3$ replicate wells per group. This quantification was performed once. **m, n** Saturation binding on HEK293A (**m**) or HEK293T (**n**) cells with stable human GPR116 OE or mock cells after incubation with the indicated concentrations of FcsFNDC4 or Fc control. For **n**, FcsFNDC4 binding was also performed in excess of sFNDC4 (1 mM) to determine non-specific binding (NSB). Calculated equilibrium binding constant (Kd) is shown for total binding in **m** and specific binding in **n**. In **m, n**, $n = 3$ replicate wells are shown of a representative experiments out of three independent experiments. In **b**, **d**, **f**–**i** (top panel), **k**, data are shown as mean ± SEM. Statistical analysis in **b**, **d**, **f**–**i** (top panel) represents unpaired two-tailed $t$ test. For **a**, **b**, **f**, **h**, **i**, **k**, **m**, **n**, a representative gating is shown in Supplementary Fig. 1a. All cells were gated and thus MFI values were derived always from all assessed cells. Source data are provided as a Source data file.

indicated certain predicted similarities for the interaction of FNDC4-GPR116, we hypothesized that FNDC4 might interact with the EC fragement of GPR116 (Supplementary Fig. 4h). Therefore, we employed a GPR116 N-terminal EC targeting antibody to investigate whether such antibody would abolish FcsFNDC4 binding in live cells. We checked the specificity of this antibody (ab111169) in 3T3L1 mature adipocytes treated with lenti-shGPR116 to induce deletion of endogenous GPR116 and lenti-shControl lentivirus treated adipocytes (Supplementary Fig. 4i, j). Indeed, in HEK293T GPR116 OE cells incubated with anti-GPR116 prior to FcsFNDC4 or Fc binding (100 nM) we observed an anti-GPR116 dose-dependent decrease in the binding of FcsFNDC4 compared to isotype control or Fc control (Fig. 4k), thus supporting that FcsFNDC4 binds to the extracellular part of GPR116.

To estimate the binding affinity of FcsFNDC4 to GPR116, we stably overexpressed human GPR116 (with C-terminal fused FLAG) in HEK293A and HEK293T cells. We confirmed equal levels of GPR116 overexpression (OE) in both cell lines by blotting against FLAG (Supplementary Fig. 4k). Of note, we measured endogenous GPR116 mRNA levels in HEK293T cells and found no detectable expression of GPR116 in those cells, as opposed to HEK293A cells (HEK293T Ct > 32) (Fig. 4l). Using the above-described fluorescent flow cytometry binding assay, we created saturation binding curves of FcsFNDC4 and Fc control to HEK cells GPR116 OE or mock controls (Fig. 4m, n). In both

HEK cell lines, only binding of FcsFNDC4 to GPR116 OE cells showed binding saturation (Fig. 4m, n). In contrast, binding of FcsFNDC4 to mock-transfected control was very low and did not saturate in HEK293A cells (Fig. 4m). In HEK293T mock-transfected cells, which express no endogenous GPR116, FcsFNDC4 binding was completely absent, similar to Fc control binding (Fig. 4n). Competition binding with excess native sFNDC4 abolished FcsFNDC4 binding, supporting specific binding of FcsFNDC4 in HEK293T GPR116 OE cells. These experiments estimated specific binding of FcsFNDC4 to GPR116 with an equilibrium dissociation constant, Kd = 33 ± 10 nM in HEK293A cells (Fig. 4m) and Kd = 25 ± 5 nM in HEK293T cells (Fig. 4n).

**GPR116 is required for the insulin-sensitizing effects of FcsFNDC4 in adipocytes**. Our data thus far suggested that the FNDC4–GPR116 axis specifically acts via liver–WAT communication. To delineate the role of AT GPR116 in the FcsFNDC4 effects on glucose homeostasis, we generated AT-specific GPR116 KO mice (GPR116$^{Ad−/−}$), using adiponectin Cre-mediated gene targeting in GPR116 flox site-carrying mice. HFD-fed GPR116$^{Ad−/−}$ and GPR116$^{Adf/f}$ mice were treated with FcsFNDC4 or Fc control for 4 weeks by ip delivery of 0.2 mg/kg every second day. Of note, FcsFNDC4 improved glucose tolerance only in GPR116$^{Adf/f}$ mice but not in the GPR116$^{Ad−/−}$ littermates, whereas Fc control injections did not have any effect in neither genotype (Fig. 5a, b).

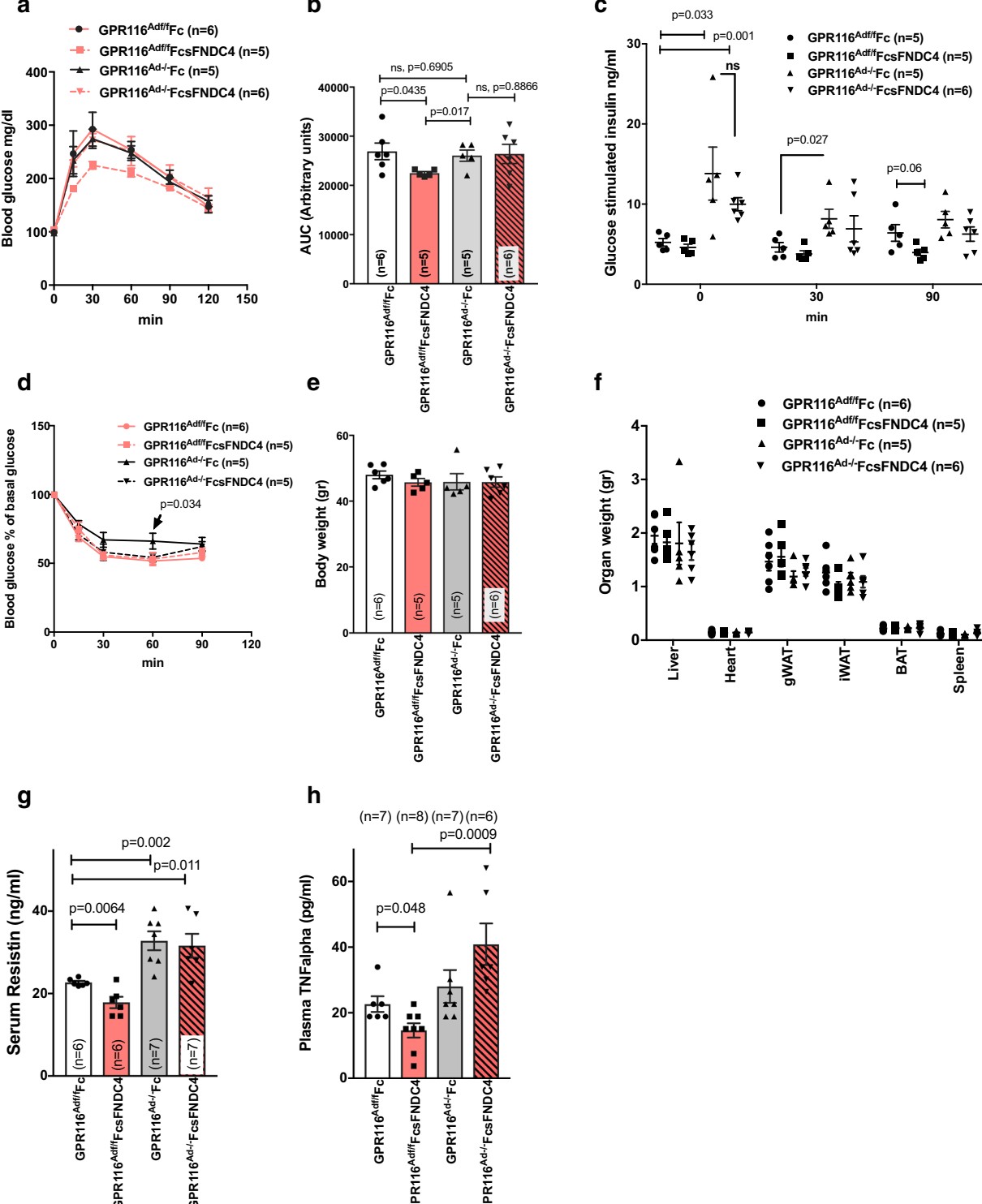

**Fig. 5 HFD fed GPR116$^{Ad−/−}$ mice did not improve glucose tolerance in response to FcsFNDC4 therapeutic injections as opposed to GPR116$^{Adf/f}$ mice.**
**a** Blood glucose during IPGTT test at the indicated time points and area under the curve (**b**). **c** Glucose-stimulated insulin response during the IPGTT in **a**, **b**.
**d** Blood glucose as percentage of baseline glucose levels during an ITT. **e** Body weight, **f** organ weight, **g** serum resistin, and **h** plasma TNFalpha at the indicated groups. White bars: *GPR116$^{Adf/f}$* Fc-injected mice, red bars: *GPR116$^{Adf/f}$* FcsFNDC4-injected mice, grey bars *GPR116$^{Ad−/−}$* Fc-injected mice, red bars/pattern: *GPR116$^{Ad−/−}$* FcsFNDC4-injected. Mice were males, set on a HFD 60% fat when 9–10 weeks old, for 12 weeks. FcsFNDC4 injections were given at a dose of 0.2 mg/kg every second day for 4 weeks (injections between weeks 8 and 12 of HFD), *n* = 6–8 mice per group. Exact number of mice are shown on figures and also described in the Source data. During IPGTT and glucose-induced insulin test, 2 g/kg ᴅ-glucose was injected and during the ITT 0.8 U/kg insulin was used ip. In all panels, data are presented as mean ± SEM. Statistics represent unpaired two-tailed *t* test. p p-value, ns non-significant. Source data are provided as a Source data file.

There were no significant differences of FcsFNDC4 on the glucose-stimulated insulin response (Fig. 5c) and insulin tolerance (Fig. 5d) in neither genotype and in comparison to the Fc-treated controls. Furthermore, we did not observe any difference of FcsFNDC4 injections on body (Fig. 5e) and organs weights (Fig. 5f) as compared to the Fc control in both genotypes. Importantly, GPR116$^{Ad-/-}$ mice on HFD demonstrated signs of pre-diabetes, manifested by fasting and glucose-stimulated compensatory hyperinsulinemia (Fig. 5c), and tended toward having higher blood glucose levels during an ITT (Fig. 5d) compared to GPR116$^{Adf/f}$. Thus, GPR116$^{Ad-/-}$ mice showed signs of pre-diabetes similar to the mice with the effects of decreased hepatic FNDC4 levels (AAVshFNDC4 mice).

To assess whether the observed anti-inflammatory effects of FcsFNDC4 were also mediated via the AT GPR116, we measured circulating resistin (Fig. 5g) and TNFalpha (Fig. 5h). FcsFNDC4 injections decreased these inflammatory markers only in GPR116$^{Adf/f}$ but not in GPR116$^{Ad-/-}$. Given that GPR116 was not expressed in macrophages and other immune cells (according to https://www.proteinatlas.org/ENSG00000069122-ADGRF5 and our own quantifications, Ct values are non-detectable), our findings supported the conclusion that adipocyte GPR116 mediated both insulin-sensitizing and local inflammation-lowering effects of FcsFNDC4.

To investigate whether FcsFNDC4 was able to improve insulin resistance directly in adipocytes and subsequently promote insulin-stimulated glucose uptake, we utilized a previously described method of in vitro induced insulin resistance in 3T3L1 mature adipocytes[14]. Exposure of 3T3L1 mature adipocytes to 10 nM insulin for 16 h (overnight (O/N)) was sufficient to induce insulin resistance reflected by dampened phosphorylation of downstream effectors of insulin receptor signaling and decreased insulin-dependent glucose uptake upon acute insulin stimulation[14]. In this paradigm, by using a wide range of FcsFNDC4 concentrations (1 pM–10 nM) (for ng/ml, see Supplementary Note 2), co-incubation of FcsFNDC4 with insulin for 16 h (O/N) was able to prevent dampening of insulin signaling due to O/N insulin exposure (Fig. 6a and Supplementary Fig. 5a). Similar observations were made for AKT substrate pAS160, which regulates GLUT4 translocation to the cell membrane[15]. This effect peaked at concentrations of 0.25 and 0.5 nM, whereas higher concentrations led to dampened signal, possibly suggesting receptor desensitization, a phenomenon typically observed in GPCR activation[16] (Fig. 6a and Supplementary Fig. 5b).

To explore the role of GPR116 in the FcsFNDC4-dependent effects on insulin-induced pAKT and pAS160, we employed an antibody, targeting the extracellular part of GPR116 (anti-GPR116) (ab111169) (Supplementary Fig. 4i, j). This antibody disrupted the binding of FcsFNDC4 to GPR116 OE HEK293T cells compared to the isotype control (Fig. 4k). Upon O/N exposure to insulin, FcsFNDC4 did not improve insulin sensitivity in the presence of anti-GPR116 antibody as it failed to enhance insulin-induced pAKT and pAS160 levels (Fig. 6b and Supplementary Fig. 5b). In addition, to exclude secondary effects of the chronic incubation, we pre-incubated healthy 3T3L1 adipocytes with GPR116-blocking antibody for 30 min prior to the addition of fresh media containing only FcsFNDC4 and insulin for 5 min. Also under these acute conditions, FcsFNDC4 enhanced insulin-induced pAKT levels; however, it failed to do so in adipocytes pre-incubated with anti-GPR116 antibody (Supplementary Fig. 5c). Importantly, under both chronic and acute conditions, FcsFNDC4–GPR116 interaction enhanced pAKT and pAS160 levels only in combination with insulin, supporting the notion that FNDC4–GPR116 does not signal directly on pAKT (Supplementary Fig. 5c, d), but acts as a necessary insulin sensitizer in WAT. Previously, it has been published that sFNDC4 promotes insulin signaling via the AMPK-

HO-1 pathway in adipocytes[6] and to also activate pSTAT3 in macrophages[4]. Therefore, we also assessed the direct effects of sFNDC4–GPR116 on pAMPK, HO-1, and pSTAT3 levels in mature adipocytes. We found no changes in pAMPK (Supplementary Fig. 5e), HO-1, or pSTAT3 upon incubation of mature adipocytes with different concentrations of FcsFNDC4 for 15 or 30 min (Supplementary Fig. 5f). Of note FcsFNDC4 demonstrated GPR116-dependent activation of pERK1/2, which is a common end point of insulin receptor activation signaling and activation readout for many GPCRs (Supplementary Fig. 5d), supporting the existence of cross-talk between the sFNDC4–GPR116 and the insulin receptor signaling.

One of the functional consequences of enhanced insulin signaling in white adipocytes is the promotion of insulin-stimulated glucose uptake via the GLUT4 transporter[15]. To investigate the role of the sFNDC4–GPR116 interaction in insulin-stimulated glucose uptake in 3T3L1 mature adipocytes, we assessed direct glucose uptake using [$^3$H]2-deoxyglucose ($^3$H-2DG). Under the above-described conditions of insulin resistance in 3T3L1 mature adipocytes, FcsFNDC4 promoted insulin-stimulated glucose uptake, which was absent in the presence of anti-GPR116 antibody (Fig. 6c). Overall, these findings underscored a functional dependence of FcsFNDC4 on GPR116 and they suggested that the interaction of FcsFNDC4 with GPR116 was required for exerting its insulin-sensitizing effects in white adipocytes.

**Interaction of FcsFNDC4 and GPR116 N-terminus induces Gs-cAMP signaling in adipocytes.** To investigate whether FcsFNDC4 triggered a typical G protein signaling via GPR116, we employed a luciferase reporter assay for G protein coupling. To that end, we generated hygromycin-resistant 3T3L1 fibroblast clonal cell lines, each carrying stable expression of transcription reporters: CRE-lu2P, cAMP response element (reporting for Gs signaling), NFAT-RE luc2P, nuclear factor of activated T-cells response element (reporting for Gq signaling), SRE-luc2P, serum response element (reporting Gαi signaling), and SRF-luc2P, serum response factor response element (reporting for G12/13 signaling)[17]. On day 8 post adipogenic differentiation of each of the above reporter cell lines into mature adipocytes, we performed dose stimulation with FcsFNDC4 or Fc control. We observed dose-dependent increase in CRE-luc2P activity 3–4 h post induction, whereas Fc control did not induce any increase in luminescence. We did not observe any change in luminescence in none of the NFAT-RE (16 h post induction), SRE (3–4 h post induction), or SRF (3–4 h post induction) reporter carrying adipocytes (Fig. 6d). As a positive control for the assay functionality, we used Forskolin 10 μM for the CRE-luc2P activity, 40% fetal bovine serum (FBS) + 20 ng/ml phorbol 12-myristate 13-acetate (PMA) for the SRE-luc2P activity and 40% FBS for the SRF-luc2P activity. For those reporters, the positive control induction resulted in a significant increase in luminescence compared to control media condition already at 3–4 h post induction; however, the induction of the NFAT-RE luc2P activity by ionomycin 1 μM + PMA 10 ng/ml required 16 h to produce a significant difference in luminescence compared to the control media. Nevertheless, stimulation with FcFNDC4 did not induce NFAT-RE luc2P activity neither after 16 h of incubation (Fig. 6d) nor after 3–4 h of stimulation (16 h stimulation is shown in Fig. 6d). These findings suggest that sFNDC4 triggers an early Gs-cAMP signaling in white adipocytes, which is consistent with the idea of targeting a GPCR receptor. To assess the dependence of this signaling on the interaction of FcsFNDC4 to GPR116 ectodomain, we performed the same dose induction in the CRE-luc2P 3T3L1 adipocytes, which were incubated with the anti-GPR116

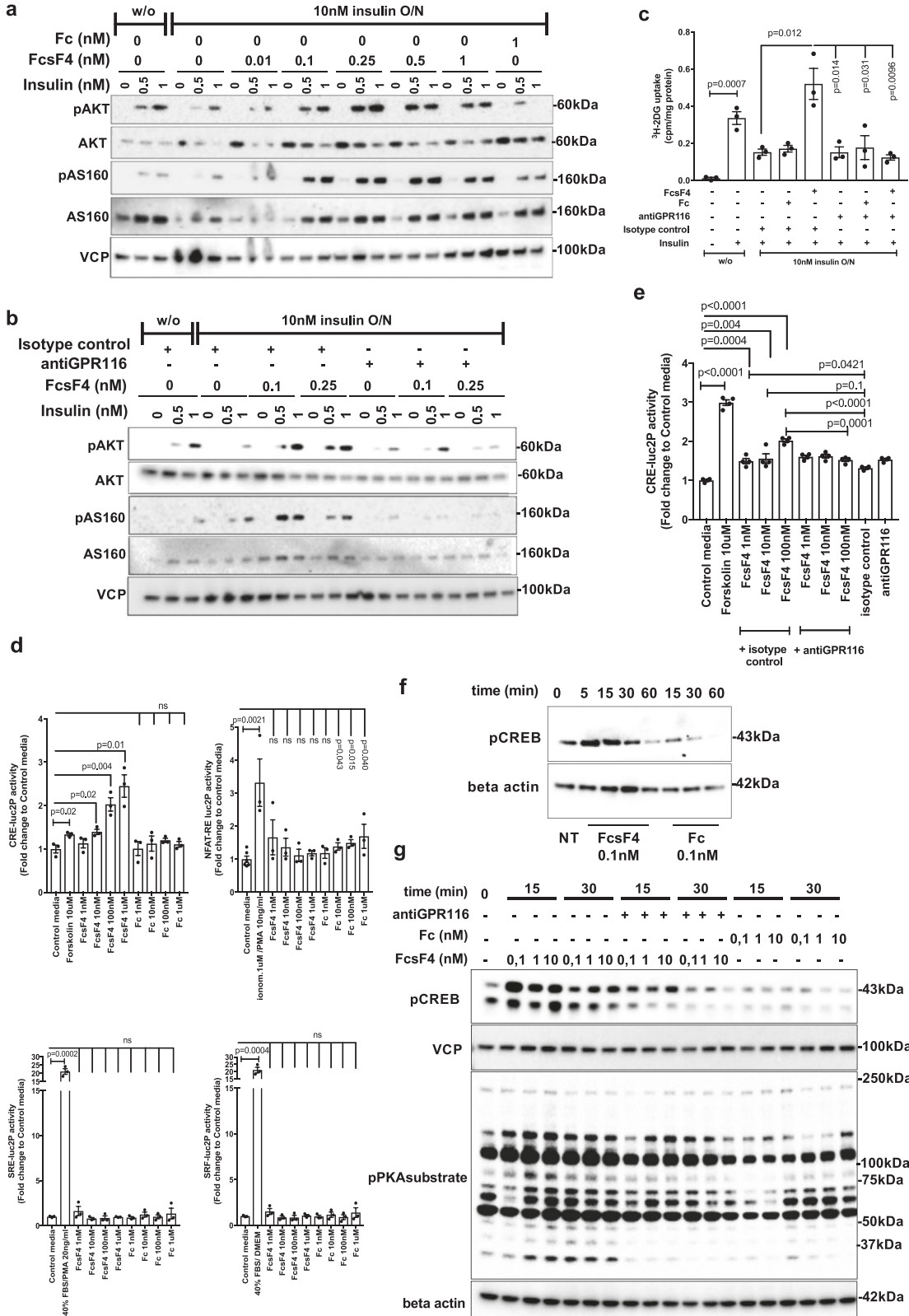

antibody, or isotype control 30 min prior stimulation with FcsFNDC4. The presence of anti-GPR116 antibody inhibited the FcsFNDC4-induced CRE-luc2P activity, whereas FcsFNDC4 indeed induced CRE-luc2P activity in isotype control-treated adipocytes (Fig. 6e and Supplementary Fig. 5g). The induction of Gs-cAMP signaling by FcsFNDC4–GPR116 was further supported by a rapid and transient induction of cAMP-sensitive

pCREB and pPKA substrate in response to FcsFNDC4 (Fig. 6f, g), which was absent in adipocytes pre-incubated with anti-GPR116 blocking antibody (Fig. 6g). Furthermore, we did not observe any changes in pPKC substrate, in response to FcsFNDC4 (Supplementary Fig. 5h). Therefore, we concluded that FcsFNDC4–GPR116 activation in white adipocytes leads to Gs coupling and activates cAMP signaling.

**Fig. 6 sFNDC4 insulin-sensitizing effects in 3T3L1 adipocytes require interaction with GPR116 and involve Gs-cAMP signaling. a** WB of the indicated proteins: overnight incubation (O/N—16 h) of 3T3L1 adipocytes with FcsFNDC4 (FcsF4) or Fc with 10 nM of insulin or without insulin (w/o). Following O/N incubations, the cells were serum starved for 3 h. After that, cells were stimulated with insulin at the indicated concentrations (0, 0.5, 1 nM) for 5 min. **b** WB of the indicated proteins: 3T3L1 adipocytes were treated overnight with insulin (10 nM) or w/o and FcsFNDC4 or Fc in the presence of antiGPR116 (ab111169) (0.4 µg/ml) or isotype control (0.4 µg/ml). Prior acute insulin stimulation for 5 min, cells were incubated in serum-free media (SFM) for 3 h. For **a**, **b**, this experiment was performed at least two times and experiment repeats are shown in Supplementary Fig. 5 and quantification of blots in Supplementary Fig. 6. **c** Tritium-labeled glucose uptake at the indicated conditions. Cells were treated as in **b** and, after serum starvation, were stimulated with 1 nM insulin for 20 min and glucose transport was initiated by the addition of $^3$H-2DG (PerkinElmer Life Sciences) (0.25 µCi/well, 50 µM unlabeled 2-deoxyglucose) for 5 min, when the experiment was terminated. $n = 3$ replicate wells of a representative experiment out of two experiments is shown. **d**, **e** Stimulation of 3T3L1 adipocytes luciferase reporter stable cells lines with the indicated concentrations of stimuli. Stimulation was 3–4 h for the CRE-, SRE-, and SRF-luc2P cell lines and 16 h for the NFAT-RE luc2P cell line. For **e**, antiGPR116 (ab111169) 0.4 µg/ml was added 30 min prior the addition of the rec. proteins then media was removed and replaced with media containing the indicated concentrations of rec. proteins. In **d**, $n = 3$ replicate wells and NFAT control $n = 6$ replicate wells of a representative experiment out of at least 3 independent experiments is shown. In **e**, $n = 4$ independent experiments. **f** WB: 3T3L1 adipocytes and **g** WB: adipocytes derived from mouse primary SVF cells were incubated in SFM for 3 h and then stimulated in SMF with the indicated dose of rec. protein or antiGPR116 antibody 0.4 µg/ml for **g** and for the indicated duration of incubation (min). For **d**, **f**, **g**, at least three independent experiments were performed. In **d**, **e**, phosphodiesterase (PDE) inhibitor, IBMX (0,5 mM), was present in the media during the treatment of Cre2LucP adipocytes, whereas in **f**, **g** no phosphodiesterase (PDE) inhibitors were present. In **c–e**, bars are mean ± SEM and statistics represents Student's unpaired two-tailed t test. p p-value, ns non-significant, NT non-treated. Phospho-antibodies: pAKT$^{Ser473}$, pCREB$^{Ser133}$, pAS160$^{Thr642}$. Source data are provided as a Source data file.

**The FNDC4–GPR116 axis responds to therapeutic interventions in diabetic mice and humans**. Finally, to investigate whether the FNDC4–GPR116 axis responded to preventive or therapeutic approaches against T2D, we examined the regulation of the ligand–receptor system in human cohorts of weight loss interventions with either bariatric surgery (BS) or diet/exercise and in mouse models of weight gain prevention using intermittent fasting or caloric restriction (CR). We found that weight loss due to BS (Fig. 7a, b) or diet/exercise (Fig. 7c, d) led to increased serum FNDC4 levels in the same patient when compared to serum sFNDC4 levels before the intervention but only in those individuals who experienced improved insulin sensitivity after the intervention (after no T2D; Fig. 7a, c). In contrast, serum sFNDC4 levels were not altered or increased in those individuals who did not improve in terms of insulin sensitivity after either intervention (Fig. 7b, d). GPR116 mRNA was also well expressed in human biopsies of subcutaneous (Subc.WAT) and visceral fat depots (Visc.WAT). GPR116 was more abundantly found in the Subc.WAT and its expression levels were affected mainly in this depot, dropping dramatically in obese subjects with IGT and IIT as well as obese with T2D (Fig. 7e). Furthermore, in fat biopsies from humans who underwent BS for weight loss, we observed, similarly to serum sFNDC4, that GPR116 mRNA levels increased after the intervention in the SC fat and showed no difference for the visceral fat (Fig. 7f, g).

Intermittent fasting and CR have been proposed as preventive dietary patterns against insulin resistance and effective weight loss methods. Intriguingly, liver mRNA levels of *Fndc4* (Fig. 7h) and gWAT *Gpr116* (Fig. 7i) as well as plasma levels of sFNDC4 (Fig. 7j) increased in mice that underwent intermittent fasting or CR compared to the ad libitum HFD-fed mice. Thus, the ligand–receptor system FNDC4–GPR116 responded to therapeutic interventions against T2D, closely correlating with the intervention-based improvements in insulin sensitivity and body weight in both mice and human patients.

## Discussion

Here, we describe an endocrine axis and its implications in pre-diabetes. We provide evidence that FNDC4 acts as an endocrine factor, which controls systemic glucose tolerance and responds to insulin-sensitizing treatments such as diet, exercise, or BS in conjunction with the reversal of T2D and improvements in insulin resistance in mice and humans. Importantly, our data provide evidence that sFNDC4 acts as a hepatokine, as we

observed that lowering of liver FNDC4 mRNA led to corresponding lowering of the sFNDC4-circulating levels in mice. Remarkably, we show that just 30% reduction of liver and circulating levels of sFNDC4 can lead to pre-diabetes manifestations, such as glucose intolerance and compensatory hyperinsulinemia. Currently, pre-diabetes in humans is targeted mainly by prescription of metformin and promotion of lifestyle changes, such as diet/exercise. However, despite those interventions many patients progress to full-blown T2D suggesting the existence of unmanaged factors. We now demonstrate here that sFNDC4 circulates in human blood and its levels negatively correlate with insulin resistance. Our data suggest that sFNDC4 levels respond very quickly to diabetogenic lifestyles, such as HFD, and do recover by successful insulin-sensitizing therapies, such as diet/exercise or BS. Although in our paired serum samples of humans exposed to HFD we observed only a 10% decrease in the circulating levels of sFNDC4, our mouse studies support that a massive drop of sFNDC4-circulating levels is not necessary to lead to pre-diabetic phenotypes. Importantly, therapeutic injections of long-lived human FcsFNDC4 effectively managed glucose intolerance in mice, providing a prototype pharmacological protein, which now is up for further optimization in future clinical trials.

The identification of the sFNDC4 receptor is a key information with respect to the in vivo mode of action of sFNDC4. Interestingly, we observed that therapeutic injections of FcsFNDC4 specifically promoted glucose uptake and insulin signaling in the WAT upon HFD. We identified the up-to-date orphan GPR116 to be a functional receptor of sFNDC4 in vitro and in vivo. Supporting the notion that FcsFNDC4 primarily targets the WAT via GPR116, FcsFNDC4 failed to improve glucose tolerance in mice lacking GPR116 specifically in AT. It is intriguing that, despite the wide expression of GPR116 in metabolic tissues, such as muscle and liver, FcsFNDC4 effects were mediated exclusively by AT GPR116. Based on the analysis of mouse tissue-specific RNA-seq data, a recent study has predicted at least 19 putative isoforms for GPR116[18]. Thus, a possible explanation, which requires experimental confirmation, could be that interaction of sFNDC4 with specific receptor isoforms may mediate the tissue-specific effects of this hormone. Most importantly, future work will focus on clarifying the relevance of human GPR116 isoforms on sFNDC4 signaling.

GPR116 is a typical member of the adhesion GPCRs family, which possess a long extracellular N terminal fragment (NTF). The NTF is self cleaved and can modulate the baseline activity of

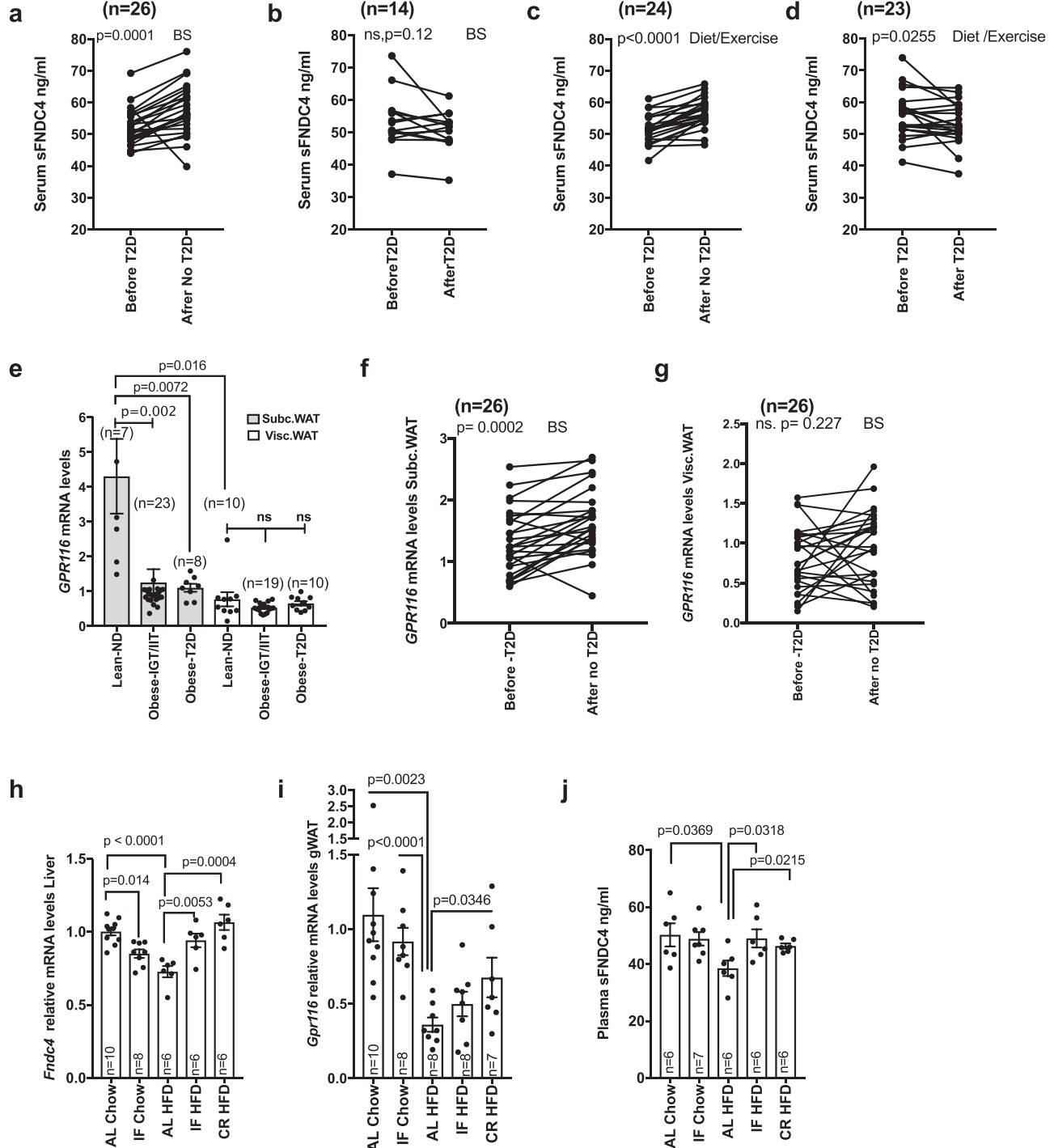

**Fig. 7 FNDC4–GPR116 axis response to anti-diabetic interventions in humans and mice positively correlates with improvements in insulin sensitivity after the intervention.** Quantification of serum sFNDC4 protein in paired samples of individuals who underwent weight loss intervention by bariatric surgery (BS) **a** ($n = 26$ humans), **b** ($n = 14$ humans), or **c** ($n = 24$ humans), **d** ($n = 23$ humans) diet and exercise. Statistics represent paired two-tailed $t$ test. **e**–**g** RT-qPCR quantification of *GPR116* mRNA at the indicated groups. ND non-diabetic, T2D type 2 diabetes, Obese-IGT/IIT: impaired glucose tolerance/insulin tolerance (cross-sectional study Leipzig), BS bariatric surgery. $n =$ human and the number of individuals is indicated on the graphs. For **e**, an unpaired two-tailed $t$ test was used. For **f**, **g**, a paired two-tailed $t$ test was used. **h**, **i** RT-qPCR quantification of *Fndc4* and *Gpr116* at the indicated tissues in mice ad libitum (AL), intermittent fasting (IF), or chow and HFD (45% fat) diet and caloric restriction (CR) upon HFD (45% fat). **j** Quantification of plasma sFNDC4 ng/ml at the indicated groups. Blood was collected from the central vein after decapitation (trunk). For **h**–**j**, $n =$ mice and the exact number of mice per group is shown on the graphs. Statistics represent an unpaired two-tailed $t$ test. In **e**, **h**–**j**, bars represent mean ± SEM. **a**–**d** and **f**, **g**: subjects' information are shown in Supplementary Table 2. For subjects' information related to **e**, see Supplementary Table 1. Source data are provided as a Source data file.

adhesion GPCRs by (a) parts of it being non-covalently associated with the extracellular interface of the 7TM part of the remaining GPCR, after NTF cleavage and (b) by interacting with other adjacent membrane or extracellular matrix proteins[19]. Signaling via the NTF of adhesion GPCRs is complex and not well understood. The current leading model of adhesion GPCRs activation is that post NTF removal, the remaining membrane bound C-terminal fragment (CTF) exposes a peptide agonist (tethered peptide), which activates the receptor in an orthosteric agonism manner to initiate interactions with heterotrimeric G proteins. NTF signaling has been shown in certain cases to act autonomously or to interplay with the CTF to initiate or not heterotrimeric G protein signaling[20]. Due to these properties, NTF may provide spatiotemporal and context-specific signaling properties to its receptor. Specifically for GPR116, it has been shown that NTF removal triggers Gq signaling and calcium mobilization via tethered peptide agonism in alveolar type II (AT2) cells[21]. However, signaling via hormone binding to the NTF-GPR116 is completely unknown. In our study, we found that sFNDC4 specifically and strongly interacted with the NTF-GPR116 and we provide evidence that this interaction led to cross-talk with the insulin signaling pathway in adipocytes, which has been shown in the past to be one mechanism via which GPCRs may affect insulin action[22]. Based on specific Cre-dependent reporter gene activation and Gs-cAMP signaling, such as pCREB and pPKA substrate, we claim that sFNDC4 via interaction with GPR116 stimulates Gs and cAMP as secondary messenger in adipocytes. Furthermore, the lack of effect of sFNDC4 on the pPKC substrate levels suggested the absence of induction of parallel Gs and Gq signaling and restricted the sFNDC4–GPR116 interaction to a specific Gs-cAMP signaling pathway in white adipocytes. Typically, Gs coupling has been linked to increased lipolysis, mitochondrial oxidation, and browning in white adipocytes[23]. In our study, we did not find any effect of FcsFNDC4 (produced in mammalian cells and as such being free of endotoxin) or native sFNDC4 (produced in Escherichia coli) on browning and mitochondrial genes of mouse primary adipocytes (Supplementary Fig. 7a) nor changes in EE after long term or acute injections of FcsFNDC4 to mice. However, opposite to potent inducers of cAMP signaling, such as isoproterenol, sFNDC4 demonstrated a rapid and transient increase in cAMP targets (pCREB, pPKA substrate). Earlier studies have highlighted the importance of acute, transient, or partial cAMP stimulation for Glut4-dependent glucose uptake[24–26], which is in line with the observed stimulatory effects of the FNDC4–GPR116 axis on adipocyte glucose uptake. It is tempting to speculate about the potential intersection nodes between the FNDC4–GPR116–cAMP and the insulin–Akt pathway. Prime candidate tracks to explore will be the AS160 signaling nodule[27], as well as a direct PKA-mediated Akt activation via physical contacts as shown in endothelial cells during late ischemic preconditioning[28]. Overall, the present study provides a paradigm on the mode of GPR116 activation and sheds light on the metabolic implications of the NTF-GPR116 in AT.

According to a previous mouse model of GPR116 deletion under the control of the Ap2Cre promoter, it has been suggested that GPR116 deletion in the AT results in systemic insulin resistance and ectopic fat accumulation[29]. The lack of specificity of the Ap2 gene expression for the adipocytes[30] prompted us to employ the Adiponectin Cre promoter to drive deletion of GPR116 in mature adipocytes. We have observed loss of the FcsFNDC4 insulin-sensitizing effects in the GPR116[Ad−/−] mice. These findings further support specific functional dependency of these two molecules also in vivo and clearly draw a liver-to-WAT endocrine axis in the control of glucose homeostasis. Furthermore, GPR116[Ad−/−] mice exhibited mild pre-diabetes symptoms, such

as compensatory hyperinsulinemia compared to GPR116[Adf/f] littermates, but these phenotypes were milder compared to those observed upon deletion of GPR116 under the control of the Ap2 promoter (Ap2CreGPR116−/− mouse). These observations may suggest that another cell type outside the adipocyte may be responsible for the severe effects of GPR116 deletion on glucose homeostasis, possibly occurring at early developmental stages.

In contrast to the low therapeutic dose of FcsFNDC4 (0.2 mg/kg) required for improvement in glucose tolerance in HFD mice in our current study, a much higher dose (3 mg/kg) has been previously utilized to treat inflammation in a mouse model of colitis[4] as well as to improve hyperlipidemia ER stress-induced insulin resistance on HFD fed mice, the later treated daily with Escherichia coli-produced native sFNDC4[6]. Of note, in our pilot work the dose of 3 mg/kg of FcsFNDC4 every other day in HFD mice had no impact on glucose homeostasis or systemic inflammation (Supplementary Fig. 7b), supporting that the effects of FcsFNDC4 on glucose homeostasis are not secondarily to the improved inflammation. At a dose of 0.2 mg/kg every other day, we observed improved glucose tolerance and improved inflammation, proposing that insulin sensitization in the adipocytes precedes the anti-inflammatory effects of FcsFNDC4. In addition, we observed a remarkable decrease of resistin in the circulation and in the WAT of HFD treated with FcsFNDC4 compared to the control mice. Resistin is a proinflammatory adipokine, which several previous studies have shown to be specifically secreted by white adipocytes in mice[8]. Due to the fact that resistin levels decrease as a response to insulin sensitization in adipocytes[31], modulation of FcsFNDC4 levels of resistin as a result of improved insulin sensitivity provide an explanation of reduced inflammation in our FcsFNDC4-injected mice secondarily to the beneficial effects of FcsFNDC4 in white adipocytes. Furthermore, the sufficiently low therapeutic dose by which sFNDC4 exerts metabolic benefits is consistent with the targeting of a high-affinity receptor in vivo and suggests that apparently a much higher dose of sFNDC4 is required to exert its anti-inflammatory effects directly on macrophages. We have found that GPR116 has almost no detectable mRNA levels in resident, induced or mouse macrophages isolated from the bone marrow, AT macrophages, or alveolar macrophages[32] (raw Ct values: >35), compared to adipocytes (raw Ct values: 25–26). However, injections of FcsFNDC4 in GPR116[Ad−/−] failed to lower circulating levels of resistin and TNFalpha, as it was the case for the FcsFNDC4 injections in GPR116[Adf/f] mice. These findings strongly support the notion that the anti-inflammatory effects of FcsFNDC4 in HFD-injected mice derive via an adipocyte GPR116-dependent mechanism. Interestingly, our present study suggest that that insulin-sensitizing mechanisms of sFNDC4-GRP116 do not involve ER stress or changes in the pAMPK-HO-1 pathway but employ Gs-cAMP signaling to prime and enhance the action of insulin in those cells. Thus, our current work clearly provides a conceptual advance not only in terms of the functions of FcsFNDC4 but also provide a targeted mechanism by which the liver–AT axis controls systemic glucose homeostasis and inflammation. In addition, our work suggests the existence of a signaling pathway in macrophages, which is perhaps of much lower specificity compared to other metabolic cells, such as adipocytes, but may have important implications for immunomodulation by sFNDC4, for example, in the case of severe and chronic inflammation, as seen in the dextran sodium sulfate colitis mouse model[4].

In conclusion, the present work uncovers a FNDC4 ligand–GPR116 receptor interaction specifically in WAT, controlling systemic glucose tolerance (Supplementary Fig. 8). Our findings suggest a potential therapeutic exploitation of the sFNDC4–GPR116 axis in human pre-diabetes as delivery of therapeutic recombinant FcsFNDC4 substantially improved

glucose tolerance in pre-clinical settings in a GPR116-dependent manner, and the sFNDC4–GPR116 axis is functionally and mechanistically conserved in humans.

## Methods

**Animals.** All mice (*Mus musculus*) were housed in a temperature-controlled (20–22 °C) room on a 12-h light/dark cycle. Mice were fed a chow diet or HFD research diets (45% fat and 60% fat) where indicated. Chow fed mice were housed 4–5 mice per cage and mice on HFD were housed 3–4 mice per cage. Experiments were performed in age- and sex-matched mice.

GPR116 KO mice have previously been described by Yang et al.[33] and Fukuzawa et al.[34] and carry deletion of exon2, which encodes for the start codon and signal peptide (SP) of GPR116. GPR116Ad−/− (adiponectinCrexGPR116 flox/flox mice) derived from breedings of the Adiponectin Cre (C57Bl6Jgenetic background) to GPR116flox/flox (C57BL6N genetic background). They were on a C57bl6N/J mixed background. In the manuscript, we used male mice 20–22 weeks old. Littermates of the same sex were randomly assigned to experimental groups. GPR116flox/flox [33] and Adiponectin-Cre transgenic mice were crossed to produce the adipose-specific GPR116 conditional KO mice: GPR116Ad−/−, which is Adiponectin Cre-positive and GPR116 flox/flox mice. Adiponectin Cre-negative GPR116flox/flox mice were used as controls. Genotyping was done according to Yang et al.[33] and the primers sequences are given in Supplementary Table 3.

WT mice for rec. protein injections, HFDs, and primary cell isolation were C57BL6N, male mice, 12–25 weeks old, and they were purchased from Charles River Laboratory. For isolation of primary SVF cells, we used WT mice, 6–7 weeks old, males, purchased from Charles Rivers Laboratory. For isolation of primary islets, WT mice, male, 8–13 weeks old, C57Bl6N, from Charles Rivers were used. For the isolation of primary SVF cells from GPR116 WT, HET, KO mice, the mice were on a C57Bl6N genetic background, males, 6–7 weeks old.

All experiments were conducted in accordance with European Directive 2010/63/EU on the protection of animals used for scientific purposes and were performed with permission from the Animal Care and Use Committee (N169/13, N412/12, N187/12 and 55.2-1-54-2532-164-2015, 55.2-1-55-2532-49-2017, 55.2-1-54-2532-125-2017, 55.2-1-54-2532-117-2016).

All mice were kept in mouse animal facility within the institute, under controlled temperature, light, and air humidity conditions. The rooms where mice were kept had ambient temperature of 20–22 °C and 46–65% relative humidity, appropriate for mice, on a 12-h light/dark cycle.

**Cell lines.** Human embryonic kidney 293 (HEK293) cells were established from female fetus. NIH3T3 fibroblasts are mouse fibroblasts, which lack adipogenic differentiation capacity to mature adipocytes. 3T3L1 are WAT mouse fibroblasts, with good adipogenic differentiation capacity. HepG2 is a human liver cancer cell line. All cell lines were cultured in Dulbecco's Modified Eagle Medium (DMEM) high glucose media with 10% FBS and 1% penicillin–streptomycin (P/S) at 37 °C in 5% $CO_2$. Immortalized SVF cells used for FACS-based receptor screening were derived from male 129SVE mice, as described in refs. [35,9]. All cell lines were purchased from ATCC.

**Primary cell cultures.** Primary islets were isolated from the pancreas of 8–13-week-old C57BL/6N mice via collagenase P (Roche) digestion as described before (Szot et al. 2007)[36] followed by a centrifugation step using an Optiprep density gradient (Sigma). Isolated islets were handpicked twice and incubated O/N in RPMI supplemented with 10% v/v FBS and 1% v/v PS for recovery.

**Study participants—human studies**
*Cross-sectional study (Leipzig)*[37]. Paired samples of subcutaneous, omental visceral AT and liver were obtained from 66 individuals (29 women, 37 men). Body Mass Index (BMI) ranged from 22.7 to 45.6 kg/m$^2$. The study was approved by the Ethics Committee of the University of Leipzig (approval numbers: 363-10-13122010 159-12-21052012) and performed in accordance with the declaration of Helsinki. All subjects gave written informed consent to use their data in anonymized form for research purposes before taking part in this study. For the study protocol, please see Wueest et al.[37].

*Weight loss intervention studies (Leipzig)*. Paired subcutaneous, omental visceral AT, and serum samples were further collected from 50 individuals (20 males and 30 females) in the context of a 2-step BS approach as described previously[38,39]. In parallel, 50 individuals underwent a combined exercise and diet intervention program over 12 months. The exercise program included 180 min per week supervised mixed strength and endurance training and a −500 kcal/day hypocaloric diet[39].

All AT samples were collected during laparoscopic abdominal surgery as described previously[40]. AT was immediately frozen in liquid nitrogen and stored at −80 °C. The study was approved by the ethics committee of the University of Leipzig (approval no: 159-12-21052012) and performed in accordance to the declaration of Helsinki. All subjects gave written informed consent before taking part in this study. For study protocol, please see Kannt et al.[39] and Schmitz et al.[38].

The categorization of patients to T2D and no T2D after the end of the intervention was done based on the following criteria: the requirement of anti-diabetic medications, hemoglobin A1c (HbA1c) <6%, and fasting plasma glucose <6.0 mmol/l.

*Measurement of body composition and metabolic parameters (Leipzig)*. BMI was calculated by weight (kg) divided by square of height (m). Weight-to-height ratio was calculated from the measured waist and hip circumference. Computed tomography or magnetic resonance imaging scans at level L4/5 were used to calculate abdominal SC and visceral fat area (cm$^2$) and dual-energy X-ray absorptiometry or bioimpedance analyses to measure body fat content (%). In a subgroup, insulin sensitivity was quantified with the glucose infusion rate during an euglycemic–hyperinsulinemic clamp as described[40,41]. All baseline blood samples were collected between 8 and 10 a.m. after an O/N fast. Plasma insulin was measured with an enzyme immunometric assay for the IMMULITE automated analyzer (Diagnostic Products Corporation, Los Angeles, CA, USA). Plasma glucose, HbA1c, high-density lipoprotein-cholesterol, low-density lipoprotein-cholesterol, free fatty acids, and TG were measured in an automated analyzer (Cobas 8000, Roche Diagnostics, Mannheim, Germany). Serum adiponectin was measured and mean subcutaneous and visceral adipocyte size was determined as previously described in Klöting et al.[40]. For subjects' information, see Supplementary Table S2.

**Analysis of *FNDC4* and *GPR116* mRNA expression in human tissues.** RNA from AT was extracted by using the RNeasy Lipid Tissue Mini Kit (Qiagen, Hilden, Germany). Quantity and integrity of RNA was monitored with NanoVue plus Spectrophotometer (GE Healthcare, Freiburg, Germany). One microgram of total RNA from subcutaneous and visceral AT and liver were reverse-transcribed with standard reagents (Life Technologies, Darmstadt, Germany). cDNA was then processed for TaqMan probe-based quantitative real-time polymerase chain reaction (qPCR) using the QuantStudio 6 Flex Real-Time PCR System (Life Technologies, Darmstadt, Germany). Expression of *FNDC4* and *GPR116* were calculated by the standard curve method and normalized to the expression of 18S rRNA (housekeeping transcript). The probes (Life Technologies, Darmstadt, Germany) for *FNDC4* (Hs01100278_g1) *GPR116* (Hs00391810_m1), and *18S rRNA* (Hs99999901_s1) span exon–exon boundaries to improve the specificity of the qPCR. For subjects' information, see also Supplementary Tables S1 and S2.

**NUtrigenomics analysis in Twins (NUGAT) study (DIfE).** The current human cohort included 92 healthy, non-obese subjects, 34 males and 58 females (46 pairs of twins—34 monozygotic and 12 dizygotic). The age ranged from 18 to 70 years with a median of 25 years. Only twins with a BMI = weight [kg]/height [m] × height [m] between 18 and 30 kg/m$^2$ were included. The dietary intervention was carried out in a sequential design and under isocaloric conditions. Individual energy requirements were calculated based on participants resting EE determined by indirect calorimetry and physical activity level assessed by questionnaire. Participants were standardized for their nutritional behavior prior to the study via a 6-week carbohydrate-rich low-fat diet (LF, 55% carbohydrate, 30% fat, 15% protein) before they switched to a 6-week HFD (40% carbohydrate, 45% fat, 15% protein) with emphasis on foods high in saturated fat[42]. Participants were given intensive, regular, and detailed dietary guidance by a nutritionist over the entire period of intervention to ensure compliance. During HFD intervention fasting glucose values and glucose tolerance were assessed by intravenous (iv) GTT as well as fasting insulin values and HOMA—insulin resistance.

The detailed study protocol has first time described in Pivovarova et al.[42]. The Pivovarova et al.[42] paper is on a subgroup of NUGAT study, the latest including in a total of 92 participants. All 92 individuals[7,43–45] of the NUGAT study have been analyzed in the present manuscript and detailed information on those subjects can be found in Frahnow et al.[43], Schüler et al.[7,44], and Schüler et al[45]. Exclusion criteria were consumptive diseases, diabetes mellitus, high-grade anemia, renal failure, moderate-to-severe heart diseases, angina pectoris, or stroke in the past 6 months, food allergy, eating disorders, body weight change ≥3 kg within 3 months prior to the study, pregnancy or breastfeeding, drugs influencing metabolic homeostasis, lipid and liver metabolism, or inflammation (e.g., systemic corticosteroids). Participants were initially screened to determine their eligibility for enrollment in the intervention study. This screening visit comprised physical examination, medical history, anthropometric measurements, and blood analysis. Additionally, a standardized 3-h, 75-g OGTT was performed. The study was approved by the Ethics Commission of Charité University Medicine Berlin. The study was conducted in the outpatient department of German Institute of Human Nutrition in accordance with the Declaration of Helsinki and registered at www.clinicaltrials.gov (NCT01631123). All subjects provided written, informed consent.

**Expression and purification of recombinant proteins.** FcsFNDC4 generation: 6xHis-Fc-sFNDC4 (FcsFNDC4) fusion protein and 6xHis-Fc (Fc) control were expressed using a pEFIRES expression vector. DNA fragment coding SP from Fndc5 fused to 6xHis-Fc was synthesized by GeneScript USA Inc. and cloned into pEFIRES modified multiple cloning site, using NheI and NotI restriction sites. The extracellular part of FNDC4 (sFNDC4) was PCR-amplified using mouse clone MR223815 (OriGene) as a template with the set of primers: forward—GAGAGC

GGCCGCTCGACCTCCCTCTCCTGTG, reverse—GAGAGAATTCATTCCCCT GTCTCTGCAATGGC. The amplified sFNDC4 fragment was cloned into a SP-6xHis-Fc-pEFIRES vector using NotI and EcoRI restriction sites to produce SP-6xHis-hFc-sFNDC4-pEFIRES expression constructs, respectively. In addition, a linker was added in between the hFc sequence and the sFNDC4 sequence: ENLTFQGAA. The final sequence was: *HHHHHH*ADLIEGRGDPKSCDKPHTCPLCPAPELLG GPSVFLFPPKPKDTLMISRTPEVTCVVVDVSHEDPEVKFNWYVDGVEVH NAKTKPREEQYNSTYRVVSVLTVLHQDWLNGKEYKCKVSNKALPAPIEK TISKAKGQPREPQVYTLPPSRDELTKNQVSLTCLVKGFYPSDIAVEWESNG QPENNYKATPPVLDSDGSFFLYSKLTVDKSRWQQGNVFSCSVMHEALH NHYTQKSLSLSPGKAAENLTFQGAARPPSPVNVTVTHLRANSATVSWDVP EGNIVIGYSISQQRQNGPGQRVIREVNTTTRACALWGLAEDSDYTVQVRSI GLRGESPPGPRVHFRTLK GSDRLPSNSSSPGDITVEGLDGERPLQTGE, with SP sequence: MPPGPCAWPPRAALRLWLGCVCFALVQAD. This construct was transfected to CHO-S cells and stable cell lines were selected using puromycin as a selection agent. For protein production, stable CHO-S suspension cultures were grown in OptiCHO medium (Life Technologies) supplemented with Ala-Glu (Sigma). Culture supernatants were loaded onto HisTrap Excel columns (GE Healthcare) in 5 mM imidazole containing Column loading buffer (1 M NaCl) with protease inhibitors, pH 7.4, washed with 5 and 20 mM imidazole containing wash buffers (0.2 M NaCl, 0.02 M NaHPO$_4$, pH 7.4) and eluted with 250 mM imidazole elution buffer (0.2 M NaCl, 0.02 M NaHPO$_4$, pH 7.4). Eluted protein was dialyzed against phosphate-buffered saline (PBS) buffer. Newly synthesized proteins were concentrated using 3 kDa Amicon Ultra-15 centrifugal units (Merck Millipore) to 1 mg/ml and snap frozen prior to the analysis. In all, 0.5 ml of the protein sample was loaded onto Superdex 10/300 GL (GE Healthcare) equilibrated in PBS buffer. Superdex 10/300 GL was calibrated using the High Molecular Weight (HMW) Kit (GE Healthcare).

**Non-Fc-fused FNDC4**. FNDC4 (aa40-160, UniProtKB-Q3TR08 Mouse FNDC4) was cloned into a pETM11 vector for bacterial expression. After expression of 1 L in TB medium, induction with IPTG and O/N growth at 20 °C, cells were collected and frozen at −80 °C until further usage. Cells were lysed in 20 mM Tris pH 8.5, 150 mM NaCl, 10 mM imidazole, 5% glycerol, and 2 mM β-mercaptoethanol and supplemented with protease inhibitor. Prior to sonication and lysate clearance by 45 min centrifugation, at 25,000 rpm, supernatant was applied to a prepacked nickel column. After washing with 20 times column volume, protein was eluted in an imidazole gradient to a final concentration of 350 mM imidazole. The His-tag version of FNDC4 was then further purified over a size exclusion chromatography column S200, whereas untagged FNDC4 was first supplemented with TEV protease, to cleave the His-tag when simultaneously dialyzed O/N against a buffer with 10 mM imidazole. Prior to gel filtration, cleaved FNDC4 domain was subjected to a nickel column to remove non-cleaved His-tag FNDC4 and His-tagged TEV protease. In both cases, the final buffer used was 10 mM Hepes, 100 mM NaCl, 5% glycerol, and 1 mM β-mercaptoethanol. Untagged FNDC4 used for the FACS binding competition assays in PBS buffer. For cell treatment experiments (islets, adipocytes), sFNDC4 (untagged) from Adipogen was used (cat.no AG-40B-0124).

**Transient OE of human RXFP1, ITGAD, and human GPR116**. Open reading frames for human RXFP1 (RC511338), ITGAD (RC224758), and human GPR116 (RC209170) were purchased from OriGene and subcloned into pENTR-CMV vector (Gateway Invitrogen). The plasmids were transfected with lipofectamine into HEK293 cells with the standard protocol. These cells were used for experiments 48 h post transfection.

**FACS binding assay and sorting**. The cells were detached by 1-min incubation in prewarmed 0.05% trypsin EDTA and additionally scraped or only scraped in ice-cold PBS. Cells were washed three times and suspended in FACS buffer (PBS with 3% FBS), by in between pelleting using centrifugation at $1000 \times g$ for 5 min at 4 °C. All steps were performed on ice and in the cold room. Fc block (1:200) was added for 20 min in FACS buffer. Recombinant proteins were then added and incubated with cells at 4 °C for 40 min, washed three times with cold FACS buffer, followed by 40-min incubation (4 °C) with anti-human IgG secondary antibody conjugated with PE (Invitrogen, H10104, 1:200). After the incubation with secondary antibody, cells were washed twice with FACS buffer and then analyzed by analytical FACS. Quantification was performed using the mean PE value within the total cell population (10,000 events were recorded). Measurements of fluorescein isothiocyanate (FITC) values were acquired to correct for cell autofluorescence. For sorting, the cells were labeled as described above and sorted for high or low mean PE and normalized to mean FITC (background fluorescence). The incubation with recombinant proteins was performed in 96-well plates with round bottom. In all, 100,000–300,000 cells were used. During incubation, cells were mixed twice by gentle vortexing. During washes, a table centrifuge for plates was used at $600 \times g$ for 2 min at 4 °C.

**FACS binding in the presence of EDTA**. The cells were detached by 1 min incubation in prewarmed 0.05% trypsin EDTA and additionally scraped in ice-cold PBS. Cells were washed three times in Krebs–Ringer buffer with the following composition: 100 mM NaCl, 5 mM KCl, 0.1 mM MgSO$_4$, 0.1 mM CaCl$_2$ 0.4 mM

K$_2$HPO$_4$, 10 mM HEPES. All steps were performed in cold. Fc block (1:200) was added for 20 min in FACS buffer and suspended. FcsFNDC4 was added at 100 nM final concentration and increasing amount of EDTA (0–10 mM). The rest of the binding protocol was performed as described above, see "FACS binding assay and sorting."

**FcsFNDC4 cell binding assay after blocking with anti-GPR116 antibody**. HEK239T cells stably overexpressing human GPR116 were detached in ice-cold PBS by scraping and pelleted by centrifugation at $1000 \times g$ 5 min at 4 °C and resuspended in FACS buffer (PBS with 3% FBS). Cells were incubated 20 min with Fc block on ice, followed by 30-min incubation with antiGPR116 or IgG isotype control at different concentrations. Antibody was removed by centrifugation at $600 \times g$ for 2 min and then 100 nM of FcsFNDC4 or Fc rec. protein were added on the cells in FACS buffer for 40 min on ice. Afterwards, cells were washed 3 times with FACS buffer at $600 \times g$ for 2 min at 4 °C. Anti-human IgG secondary antibody conjugated with PE (Invitrogen, H10104, 1:200) was added for 40 min incubation (4 °C). Final two washes with FACS buffer and centrifugation at $600 \times g$ for 2 min at 4 °C were performed in order to remove excess PE-conjugated antibody.

**Transcriptomics**. For identifying differentially expressed genes between HBC and LBC, Affymetrix mouse Chips 2.0St arrays were performed in HBC and LBC and differentially expressed genes were selected based on p value <0.05, calculated using Student's t test and false discovery rate analysis. Three technical replicates were used and genes were selected on the basis of mean probe intensity >100 in both groups. Affymetrix global gene expression analysis from sorted LBC and HBC was performed in the BEA core facility at the Karolinska Institute. Microarray data are available in GEO (gene expression omnibus) under the number: GSE165329.

**GPR116–FNDC4-binding experiments**
*Pull down*. In all, 30 μg of 6xHisFcsFNDC4 or 6xHisFc was bound to 2 mg of anti6xHis Dynabeads (Invitrogen) for 25 min at room temperature (RT), under rotation, in 1× binding buffer/wash buffer prepared according to the supplier's protocol. Supernatant was collected and the beads were washed 4× for 2 min each wash, under rotation with 1× binding/wash buffer. A 100-cm$^2$ dish, with confluent NIH3T3 cells was lysed in 1× pull-down buffer (as described by the supplier) supplemented with 1% TritonX-100. Cells were scraped and passed through a 25-G syringe 10 times and a 27-G syringe for an additional 10 times, on ice. Finally, cell lysates were sonicated 2× for 10 min at 4 °C in a water bath sonicator and the supernatant was collected after 10-min centrifugation at 10,000 rpm, 4 °C. Protein was quantified by the BCA Pierce Assay Kit (Invitrogen), and 300 μg of supernatant was added on recombinant protein pre-bound beads for 30 min at RT. The supernatant containing un-precipitated proteins was collected and beads were washed 4× for 2 min each time, under rotation with 1× binding/wash buffer. Immunoprecipitated proteins were eluted in 100 μl His elution buffer, as described by the supplier, for 15 min, at RT, under vigorous shaking. Samples were reduced in β-mercaptoethanol containing sample buffer and boiled (98 °C) for 7 min. Ten microliters from each sample was loaded on 7.5% TGX premade mini gel from BIO-RAD, and protein was transferred to a polyvinylidene difluoride (PVDF) membrane with semi-dry transfer, under constant voltage of 10 V for 30 min, using the Trans-Blot Turbo transfer system from BIO-RAD. Membranes were blotted against anti-GPR116, using the anti-GPR116 antibody ab136262, from Abcam.

**Lentivirus packaging, infection, and stable HEK293 clone cell selection**. The lentivirus-based expression vector was also constructed using the Gateway system (Invitrogen). Human GPR116 with a C terminal FLAG tag was recombined to plenti6/V5 DEST vector from the pEntry1a GPR116 plasmid. The purified plasmid was then transfected to HEK293FT cells together with packaging plasmids from Invitrogen (ViraPower™ Lentiviral Packaging Mix). Seventy-two hours later, the supernatant of the transfected cells containing lentiviral particles was harvested and was used to infect new cells. Seventy-two hours after infection with lentivirus particles, cells were exposed to blasticidin (Invitrogen) for stable selection. Medium was changed every 3 days with fresh blasticidin. Two weeks later, the cells were trypsinized and re-suspended as single-cell suspension into 96-well plates. Thereafter, single-cell clones were amplified.

**Construction of shGpr116 lentivirus and transduction of pre-adipocytes and adipocytes**. Short hairpin RNA (shRNA) lentiviral plasmids (pGFP-C-shlenti) against mouse Gpr116 were purchased from Origene (CAT#: TL517926), and four 29mer shRNA sequences were used for silencing mouse Gpr116 TL517926A (TL517926A) 5'-tactccattcacaccactgtcatcaacaa-3' (SEQ ID NO.: 13), TL517926B (TL517926B) 5'-tcgcagtgttctgccacttcaccaatgca-3' (SEQ ID NO.: 14), TL517926C (TL517926C) 5'-cgtcatcttagacaagtctgccttgaact-3' (SEQ ID NO.: 12), and TL517926D (TL517926D) 5'-tgtggctggtgctatccacgacggtcgct-3' (SEQ ID NO.: 15), and a non-effective 29-mer scrambled shRNA cassette in pGFP-C-shLenti Vector, (CAT#: TR30021) 5'-gcactaccagagctaactcagatagtact-3' (SEQ ID NO.: 16) was used as a control. For virus production, shRNA lenti vectors were cotransfected O/N, with packaging plasmids psPAX2 (Addgene) and PMD2.G (Addgene) to HEK293FT cells, using Lipofectamine 3000. Twenty-four hours later, media was changed by DMEM 10% FBS containing 1% BSA. After 24 h, the supernatant was

recovered, filtered with 0.45 mm filters, and used to infect differentiated mature primary adipocytes. One milliliter of the supernatant was added in 1 well of a 12-well plate for 24 h. After that, the media was changed to complete DMEM media. GPR116 mRNA levels were more than 70% deleted, compared to the scrambled control and it was achieved as early as 72 h post infection.

**Reporter luciferase gene assays on 3T3L1 stable reporter cell lines.** CRE-, NFAT-RE, SRE-, and SRF-luc2P transcription activity luciferase reporters from Promega, cat. no. E8471, E8481, E1340, and E1350, respectively, were transfected with Lipofectamine 3000 to 3T3L1 fibroblasts (passage 12). Seventy-two hours post transfection, the media was changed into media with 200µg/ml hygromycin B (Invitrogen cat. no 10687010) and they were selected for 10–12 days. Clonal cell lines were generated with the limited dilution method. Media was DMEM + 10% FBS + 1% P/S + 200 ng/ml hygromycin B and was refreshed every 2 days. For the experiments, cells were differentiated into mature adipocytes with the protocol described before[46]. After the stimulation as described in the main text, cells were lysed with 1× Reporter lysis buffer—Promega cat. no. E3971, and luminescence was read after the addition of Steady-Glo® Luciferase Assay System – Promega cat. no. E2520 with the Varioscan Lux plate reader, in a white 96-well plate.

Experiments were performed at day 8 post differentiation. For the CRE-, SRE-, and SRF-luc2P, prior to stimulation with recombinant FcsFNDC4 and Fc, the adipocytes were serum starved in DMEM high glucose for 4–5 h. Stimulation was performed in serum-free conditions for additional 3–4 h. For the CRE-luc2P, all conditions included 0.5 mM 3-isobutyl-1-methylxanthine (IBMX) (Sigma: I5879). For the NFAT-RE luc2P, stimulation was performed in 10% FBS + DMEM high glucose for 16 h. As positive controls for assay functionality, the following were used: Forskolin 10 µM (Cay11018-1, Biomol), ionomycin 1 µM (sc-3592, Santa Cruz), and PMA 10–20 ng/ml.

For the stimulation of Cre-luc2P reporter adipocytes, phosphodiesterase inhibitor IBMX (0.5 mM) was present during the stimulation with rec. proteins or Foskolin.

**In vivo insulin signaling studies.** On HFD-fed mice: Mice were fasted O/N (12–16 h) and subsequently were injected (ip) with 5 U/kg Humulin and organs were excised after 8 min and snapped frozen in liquid nitrogen.

**AAV knockdown in mice.** AAV8-U6-GFP-scrmb-shRNA and AAV8-U6-GFP-shFNDC4 were purchased from Vector Biolabs and injected iv. to 9–10-week-old mice at $1 \times 10^{12}$ GC per mouse. Mice were given a HFD at 10–11 weeks of age. HFD was 45% fat D12451, Research Diets.

**Islet isolation and glucose-stimulated insulin secretion assay (GSIS).** Primary islets were isolated from the pancreas of 8–13-week-old C57BL/6N male mice via collagenase P (Roche) digestion as described above, followed by a centrifugation step using Optiprep density gradient (Sigma). Isolated islets were handpicked twice and incubated in RPMI supplemented with 10 % v/v FBS and 1 % v/v P/S O/N for recovery. The next day, islets were treated with various concentrations of the commercially available bacterial FNDC4 (Adipogen), the in-house produced mammalian FcsFNDC4, or the corresponding negative controls PBS and Fc-peptide for 24 h. To gain sufficient islet material, islets of two mice were pooled for each biological replicate. For the GSIS assay, 9 islets of comparable size were transferred per well into a low-attachment V-shaped 96-well plate. Islets were incubated in modified Krebs Ringer phosphate HEPES buffer (KRPH; 115 mM NaCl, 4.7 mM KCl, 1.2 mM $KH_2PO_4$, 1.2 mM $MgSO_4 \cdot 7H_2O$, 20 mM $NaHCO_3$ 20 mM, 16 mM HEPES, 2.56 mM $CaCl_2 \cdot 2H_2O$) supplemented with 0.1 % BSA (RIA grade) with various glucose concentrations in the presence of the proteins described above. Exendin-4 served as positive control. After incubation in the presence of 1 mM glucose for 1 h, islets were sequentially incubated with 2.8 mM glucose (low glucose) and 16.7 mM glucose (high glucose) for 30 min each. In between the incubation steps, islets were washed twice using KRPH with 2.8 mM glucose. Insulin concentration in the supernatant was assessed using the mouse insulin ELISA kit from ALPCO.

**C-peptide and insulin measurements.** For measurements of C-peptide in plasma, C-peptide quantification kit from CrystalChem was used (cat.# 90050) and for measuring C-peptide in serum the C-peptide quantification kit from ALPCO (cat. no.: 80-CPTMS-E01) was used, according to the manufacturer's description. Insulin was measured with a commercial kit from ALPCO (cat. no.: 80-INSHU-E01.1).

**Cytokine and adipokine ELISA.** ELISA quantification of TNFalpha, leptin, adiponectin, and resistin was performed according to the kit's instructions—R&D Systems.

**Therapeutic injections of FcsFNDC4 to HFD (60% fat) mice.** WT C57BL6N, male mice were fed on a HFD with 60% fat (Research Diets Cat. # D12492) for 16 weeks, starting from 8 to 9 weeks of age. Mice were given ip injections of FcsFNDC4 (0.2 mg/kg) or vehicle control (PBS) every second day for 4 weeks,

while mice still on HFD (60% fat). Glucose clearance and insulin tolerance was assessed with an IPGTT and ip ITT.

2-NBDG tissue uptake quantification: Male C57BL6N mice were on HFD (45% fat) for 20 weeks and in the last weeks (16–20 weeks) were injected ip every other days with FcsFNDC4 (0.2 mg/kg) or vehicle control. Prior the IPGTT with fluorescent glucose (2-NBDG), mice were fasted for 6 h and were injected ip with a mix of normal D-glucose (2g/kg, 40%v/v) with 2% 2-NBDG glucose (stock concentration 5 mg/ml—ThermoFisher, cat.# N13195). Mice were sacrificed at 35 min after injection, which based on pilot studies was the time point 5 min after all mice had shown a peak in blood glucose (peak was at 30 min). Tissues were collected and snap frozen in liquid N. Tissues were weighed and homogenized in RIPA buffer and fluorescence was measured in a plate reader.

**Glucose and ITT.** Before testing mice were placed in a new cage and food was removed for 6 h. After this period of fasting, we assessed glucose tolerance by ip injection of 2 g/kg D-glucose at time point 0 min and subsequent measurements of blood glucose at 0, 15, 30, 60, 90, 120, and 180 min using ACCU-CHEK glucometer strips. To measure glucose-induced insulin secretion, we collected blood at 0, 15, 30, and 90 min in EDTA-coated tubes. Plasma was collected after spinning the blood at $2000 \times g$ for 10 min. To assess insulin tolerance, we measured blood glucose levels at several times points after ip injection of insulin. Insulin used was Humulin.

**Histology.** Liver and AT tissue samples were fixed in neutrally buffered 4% formaldehyde solution for 24 h (Formalin 10% neutral buffered, HT501128, Sigma-Aldrich, Germany) and subsequently routinely embedded in paraffin (Tissue Tec VIP.5, Sakura Europe, Netherlands). Sections of 3 µm nominal thickness were stained with hematoxylin and eosin (H&E), using a HistoCore SPECTRA ST automated slide stainer (Leica, Germany) with prefabricated staining reagents (Histocore Spectra H&E Stain System S1, Leica, Germany), according to the manufacturer's instructions. Histopathological examination was performed by a pathologist in a blinded fashion (i.e., without knowledge of the treatment–group affiliations of the examined slides). Immunohistochemical (IHC) detection of CD68 in eWAT and iWAT sections was performed on a Ventana Discovery Ultrastainer (Roche Diagnostics, Germany), using specific antibodies (polyclonal rabbit anti-CD68 antibody, #125212, Abcam, USA, and secondary antibody: goat anti-rabbit IgG antibody (H + L), biotinylated, BA-1000, Vector, Germany) and prefabricated solutions (DISCOVERY DAB Map Detection Kit, Cat. 760-124, Roche, USA). All IHC analyses included appropriate negative control slides (omission of the first antibody). H&E-stained slides and IHC sections were digitally scanned with an Axio Scan.Z1 scanner (Zeiss, Germany), using a ×20 objective. Automated digital image analysis (Definiens Developer XD 2, Definiens AG, Germany) was used for determination of the mean adipocyte section profile areas, as well as the numbers of CD68-positive macrophage cell section profiles and the percentage of CD68-positive stained area per total AT section area.

**Tissue lipid extraction and TG measurements.** Lipid were extracted according to Folch method[47]. Briefly 10–100 mg of frozen wet tissue were weighed, to which 1.5 ml of chloroform:methanol (2:1) mixture (maintained at −80 °C; final volume is about 1.6 ml) were added. Tissues were lysed with the Qiagen TissueLyser (2 × 30 s, 30 Hz) until no visible large particles remained. The lysed solution was spun down briefly and mixed for 20 min on Thermomixer at 1400 rpm, RT and centrifuged for 30 min at 13,000 rpm at 20 °C. Afterwards, 1 ml of supernatant (i.e., liquid phase) was transferred to a new 2-ml tube and 200 µl of 150 mM (0.9%) NaCl was added and mixed by vigorous shaking and centrifuged for 5 min at 2000 rpm. The resulted lower organic phase was transferred into new tube containing the chloroform:Triton-X (40 µl of chloroform:Triton-X (1:1) solution). This solution was dried with the speedvac O/N (or until no change in tube weight) and the remaining triton–lipid solution was resuspended in 200 µl of $dH_2O$ and nixed by 1 h rotation at RT and then stored at −80 °C until use (final volume 225 µl, 1.125 dilution factor). TGs were measured by the Sigma Triglyceride determination Kit, Cat. # TR0100.

**Tritium 2-deoxyglucose uptake assay.** 3T3-L1 adipocytes in 12-well plates were washed twice and incubated with serum- and bicarbonate-free DMEM containing 20 mM HEPES, pH 7.4, and 0.2% BSA for 2 h. Following 3-h serum starvation, cells were washed twice with Krebs–Ringer phosphate buffer (0.6 mM $Na_2HPO_4$, 0.4 mM $NaH_2PO_4$, 120 mM NaCl, 6 mM KCl, 1 mM $CaCl_2$, 1.2 mM $MgSO_4$, 12.5 mM HEPES, pH 7.4) supplemented with 0.2% BSA. Indicated dose of insulin was added for 20 min, and glucose transport was initiated by the addition of $[^3H]$-2DG (PerkinElmer Life Sciences) (0.25 µCi/well, 50 µM unlabeled 2-deoxyglucose) for 5 min. To determine non-specific glucose uptake, 25 µM cytochalasin B was added prior to the addition of $[^3H]$-2DG. Uptake was terminated with three rapid washes in ice-cold PBS, after which the cells were solubilized in 1% Triton X-100 in PBS. Samples were assessed for radioactivity by scintillation counting. Each condition was performed in triplicate.

For calculation of uptake, non-specific uptake was subtracted by the specific uptake and values were normalized for protein concentration.

**Table 1 Antibody ID and working dilutions.**

| Antibodies | Supplier | Cat. no. | Working dilution, application |
|---|---|---|---|
| Anti-P-ERK1/2 | Cell Signaling Technology | Cat.# 9101 | 1:1000 WB |
| Anti-ERK1/2 | Cell Signaling Technology | Cat. # 4695 | 1:1000 WB |
| Anti-GPR116 | Abcam | Cat.# ab136262 | 1:1000 WB |
| Anti-GPR116 | Abcam | Cat.# ab111169 | 1:1000 WB 0.4 µg/ml as a blocking antibody in vitro |
| Anti-FNDC4 | Sigma Aldrich | Cat. # SAB1401807 | 1:1000 WB |
| Anti-beta tubulin | Sigma Aldrich | Cat. # T5201 | 1:1000 WB |
| Anti-VCP | Abcam | Cat. # ab109240 | 1:1000 WB |
| Monoclonal ANTI-FLAG® M2-Peroxidase (HRP) | Sigma Aldrich | Cat. # A8592 | 1:1000 WB |
| Anti-rabbit, HRP conjugated | DAKO | Cat.# P0399 | 1:10,000 WB |
| Anti-mouse IgG-HRP conjugated | DAKO | Cat.# P0447 | 1:10,000 WB |
| Anti-F(ab')2-goat anti-human IgG Fc secondary antibody, PE | Invitrogen | Cat.# H10104 | 1:200 FACS |
| Anti- Cd68 | Abcam | Cat.# ab125212 | 1:500 IHC |
| Anti-beta actin | Sigma Aldrich | Cat.# A5441 | 1:1000 WB |
| Anti-pAS160(Thr642) | Cell Signaling Technology | Cat.# 4288 | 1:1000 WB |
| Anti-AS160 | Cell Signaling Technology | Cat.# 2670S | 1:1000 WB |
| Anti-pAKT(Ser473) | Cell Signaling Technology | Cat.# 9271S | 1:1000 WB |
| Anti-AKT | Cell Signaling Technology | Cat.# 9272 | 1:1000 WB |
| Anti-phopho-PKA substrate | Cell Signaling Technology | Cat.# 9624 | 1:1000 WB |
| Anti-pCREB(Ser133) | Cell Signaling Technology | Cat.# 9198 | 1:1000 WB |
| Anti-pAMPKa(Thr172) | Cell Signaling Technology | Cat.# 2535 | 1:1000 WB |
| Anti-HO-1 | Cell Signaling Technology | Cat.# 43966 | 1:1000 WB |
| Anti-pSTAT3(Tyr705) | Cell Signaling Technology | Cat.# 9145 | 1:1000 WB |
| Anti-STAT3 | Cell Signaling Technology | Cat.# 12640 | 1:1000 WB |
| Anti-phopsho-PKC substrate | Cell Signaling Technology | Cat.# 6967 | 1:1000 WB |
| Secondary antibody: goat anti rabbit-biotinylated | Vector | BA-1000 | 1:750 IHC |
| Rabbit IgG, polyclonal—isotype control (ChIP grade) | Abcam | Cat.# 171870 | 0.4 µg/ml as a blocking antibody in vitro |

## FNDC4 signaling in 3T3L1 adipocytes and primary mouse SVF-derived adipocytes

*3T3L1 differentiation protocol.* Differentiation of 3T3L1 to mature adipocytes was done according to the protocol by Zebish et al.[46]. Experiments were performed of days 8–12 of differentiation and passage numbers between 15 and 20.

*Induction of in vitro insulin resistance and long-term incubation with FcsFNDC4.* Insulin resistance was induced on 3T3L1 mature adipocytes by O/N (16 h) exposure to 10 nM insulin, in DMEM high glucose, 10% FBS, 1% P/S (complete media), according to the protocol of Tan et al.[14]. Different concentrations of FcsFNDC4 or Fc controls were added to the cells for 16 h together with insulin (10 nM), with or without anti-GPR116 (ab111169) or isotype control. Also cells without insulin were included (w/o), as control for the 16 h insulin effect. After 16-h incubation described above, media was removed and cells were washed twice with PBS. Cells were incubated in serum-free high glucose DMEM for 3 h and then fresh serum-free high glucose DMEM containing different concentrations of insulin (0nM, 0.5nM, 1 nM) were added to the cells for 5 min. At that time, media was removed and cells were washed twice with ice-cold PBS and lysed in 1x cell lysis buffer from Cell Signaling, Cat. # 9803 with additional protease and phosphatase inhibitors.

For the short-term co-stimulation with FcsFNDC4 and insulin: 3T3L1 mature adipocytes were incubated for 3 h in serum-free, high glucose DMEM. 30 min before the co-stimulation with insulin and FcsFNDC4, anti-GPR116 (ab111169) or isotype control was added to the media. Then media was changed to fresh serum-free, high glucose DMEM with insulin and FcsFNDC4 for 5 min.

*Differentiation of mouse primary SVF to mature adipocytes.* Primary SVF cells were isolated by 8 min collagenase II (Life Technologies, Cat # 17101015) digestion of inguinal WAT (after removal of the lymph node) from male mice, 6–8 weeks old. SVF preadipocytes were grown till confluence in DMEM high glucose, 10% FBS + 1% P/S, when differentiation was initiated with the addition of dexamethasone (2 ng/ml), IBMX (122 ng/ml), T3 (6.7 ng/ml), insulin (0.865 ng/ml) and Rosiglitazone (5 ng/ml) for 2 days. On day 3, media was changed to complete media containing 0.865 ng/ml insulin for 2 days, and on day 5, the media was changed in complete media containing 0.4325 ng/ml insulin, in which they were maintained. Experiments were performed between day 6 and day 8 post differentiation, in serum-free, DMEM high glucose media, after 3 hr starvation in serum-free, DMEM high glucose media.

## Incubation of 3T3L1 with palmitate.
Sodium palmitate (Sigma) was conjugated with 2% BSA (fatty acid free), dissolved in DMEM, and applied to the adipocytes to a final concentration of 200 µM for 24 h with or without FcsFNDC4 (200 nM) or Fc (200 nM) control.

**Western blot**. Soluble lysates were collected by centrifugation at $15,000 \times g$ for 15 min. Samples were boiled for 7 min in Laemmli buffer and loaded to a Tris-glycine gel. For receptor blots, samples were warmed at 70 °C for 30 min and not boiled. Proteins were transferred to a PVDF membrane with semi-dry transfer for 14 min at constant 1.4 mA using the Trans-Blot Turbo transfer system from BIO-RAD. Membranes were blocked with 5% milk for 30 min and incubated O/N with the primary antibody (1:1000) in 2% milk. Washes were done with 0.1% and 0.05% TBS-Tween buffer and secondary antibody was used 1:1000 for 1 h at RT. Antibodies used in this study are listed below (Table 1). Membranes were stripped and reblotted. In all blots presented in the manuscript, loading control has been run on the same blot as the protein assessed.

**Metabolic phenotyping (TSE phenotyping)**. EE, locomotor activity, RER, and food intake were measured by combined indirect calorimetry over 93.4 h (PhenoMaster; TSE Systems, Bad Homburg vor der Höhe, Germany) as described previously[48].

Chronically injected mice with FcsFNDC4 were WT C57BL6N, male mice fed on a HFD with 45% fat (Cat. # D12451) for 16 weeks, starting from 8 to 9 weeks of age. Mice were given i.p. injections of FcsFNDC4 (0.2 mg/kg) or Fc negative control (0.2 mg/kg) every second day for 4 weeks while mice still on HFD (60% fat).

Single-dose injection of FcsFNDC4: Young male mice, 12–14 weeks old were put in the PhenoMaster for 4 days to acclimatize. On the day of the experiment, food was removed 4 h before the injection of rec. protein, when FcsFNDC4 or Fc control, 1 mg/kg, were injected i.p. to the mice. Measurement of EE, RER, and locomotor activity were collected for 2 h post injection, when the experiment was terminated.

**Altered feeding experiments in mice**. All mice were maintained on a daily cycle of 12-h light (06:00–18:00) and 12-h darkness (18:00–06:00). For CR: ad libitum groups were allowed free access to either standard chow (AL chow) or HFD 45% fat (D12451, Research Diets) for 17 weeks, while the food intake of CR groups (CR chow and CR HFD) was reduced by ~11% for 17 weeks, when the animals were sacrificed and organs harvested. For intermittent fasting: mice from 6 weeks of age were put under cycles of 24 h fasting followed by 48 h refeeding. During the fasting, only access to water was available and during the refeeding mice had ad libitum access to food. All groups were maintained on this feeding cycle either on chow or HFD 45% fat (D12451, Research Diets) for 17 weeks. Blood was collected from the trunk after decapitation.

**Real-time quantitative PCR**. mRNA was extracted with the TRIzol reagent (Invitrogen, ThermoFisher Scientific). For mouse and cell experiments, mRNA 500 ng–1 μg of RNA was amplified with the RevertAid First Strand cDNA Synthesis Kit Cat. # K1622. cDNA was PCR amplified with TaqMan Gene Expression Master Mix Cat. # 4369016 and PowerUp SYBR Green Master Mix 5x, Cat. # A25777. qPCR primers were designed to span exon–exon sequences to generate a product of 100–200 bp and sequences were derived either from the validated Primerbank (http://pga.mgh.harvard.edu/primerbank) or from the published literature. The sequence of primers and Taqman probe identity is given in Supplementary Table 3. The mRNA levels of each gene was calculated with the 2^ddCt method and normalized for the expression mRNA of the housekeeping gene (as indicated in the figure legends). We used Applied Biosystems QuantStudio 6 and 7 Flex Real-Time PCR, ThermoFisher.

**Quantification and statistical analysis**. All values in graphs are presented as mean ± SEM. Two-way analysis of variance for multiple comparison were used to analyze the data. Significant differences between two groups were evaluated using a two-tailed, unpaired or paired Student's $t$ test as the sample groups displayed a normal distribution and comparable variance (*$p < 0.05$, **$p < 0.01$, ***$p < 0.001$). In most figures exact p-values are shown.

For quantification of western blots, we performed band densitometric analysis (in Supplementary Fig. S7 are the quantification of band intensities is shown), using Image Lab version 5.2.1 build 11 (BIO-RAD).

**Reproducibility**. We repeated key experiments several times and the time of repetitions for each experiment with same results is mentioned in the figure legends.

**Reporting summary**. Further information on research design is available in the Nature Research Reporting Summary linked to this article.

## Data availability

Source data are provided with this paper. Mouse models are available with MTA agreement. All data supporting the findings within the article and its supplementary information files are available from the corresponding authors upon request. Microarray data are available in GEO under the number: GSE165329. pTPM data for FNDC4 expression are available from Protein Atlas and exact URLs for each tissue are included in the Source data file.

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

## Acknowledgements
We thank Dr. Brad St. Croix (Tumor Angiogenesis Section, MCGP, NCI at Frederick, NIH, Frederick, MD 21702, USA) for generously providing the GPR116KO and GPR116flox/flox mice. The authors would like to thank Venetia Bazioti, Thomas Malmborg, Dr. Donna Maretta Ariestanti, Hikaru Ando, Malin Elvén, Sabine Hartig, Andrea Takacs, and Elena Sophie Vogl for technical assistance with experiments. We also thank Dr. David Brodin for support with the preparation, analysis and submission of the Affymetrix microarrays data. This work was supported by grants of the Deutsche Forschungsgemeinschaft, Obesity Mechanisms (SFB 1052, B01 to M.Blüher), Karolinska Institute derived funds (ERC 309596), the Helmholtz Association (Cross-Program Topic AMPro to S. Herzig), the DFG TRR 152 and DFG TRR 296 to Timo D.Müller. The NUGAT study was funded by the German Federal Ministry of Education and Research (BMBF; grant no. 0315424) to A.F.H.P. The Rubicon Netherlands Organization for Scientific research (NWO)-mobility grant and the EFSD grant supported A. Georgiadi.

## Author contributions
A. Georgiadi: conceived the study, performed the majority of the experiments and data analysis and wrote the manuscript with help and comments from all authors. A. Georgiadi, V.L.S., R.E.M., R.A.K., A.M., X.M., K. Klepac, L.B., A.J.A., M. Bosma, O.S., O.R., M.L., A.L., D.T., I.K., K.F., T.D.M., J.M.: collected experimental data and assisted with experiments. L.B.: performed the isolated primary pancreatic islets experiments. A.M.: purified recombinant untagged Fndc4. K.K.: performed the intermittent fasting study on mice. P.N., A. Geerlof, M. Sattler, M. Scheideler, R.T., T.S.: assisted with data analysis and interpretation. A.B. and A.F.: performed the histological examination. K. Kessler, S. Hornemann, M.K., O.P.R., A.F.H.P: provided human serum from the NUGAT study and contributed to data analysis and interpretation. N.N., S. Hirose: provided preliminary data on GPR116KO mice that led to further experiments. A.D., M. Blüher: provided human serum from the Leipzig cross-sectional and intervention studies (BS, Diet/Exercise) and contributed to data analysis and interpretation. S. Herzig: conceptually supervised the study and wrote the manuscript together with A. Georgiadi.

## Funding

## Competing interests
The authors declare no competing interests.

## Additional information

[1]Institute for Diabetes and Cancer, Helmholtz Diabetes Center, Helmholtz Centre Munich, German Research Center for Environmental Health, Neuherberg, Germany. [2]Joint Heidelberg-IDC Transnational Diabetes Program, Heidelberg University Hospital, Heidelberg, Germany. [3]Chair Molecular Metabolic Control, Medical Faculty, Technical University Munich, Munich, Germany. [4]German Center for Diabetes Research (DZD), Neuherberg, Germany. [5]Department for Cell and Molecular Biology, Karolinska Institute, Stockholm, Sweden. [6]Institute of Structural Biology, Helmholtz Centre Munich, German Research Center for Environmental Health, Neuherberg, Germany. [7]Rudolf-Schönheimer-Institute for Biochemistry, Faculty of Medicine, University of Leipzig, Leipzig, Germany. [8]Biomedicum Helsinki and Department of Bacteriology, Haartman Institute, University of Helsinki, Helsinki, Finland. [9]Department of Life Science and Technology, Tokyo Institute of Technology, Yokohama, Japan. [10]Institute of Experimental Genetics, Helmholtz Diabetes Center, Helmholtz Centre Munich, German Research Center for Environmental Health, Neuherberg, Germany. [11]Department of Surgery, University of Leipzig, Leipzig, Germany. [12]Core Facility Pathology & Tissue Analytics Research Unit Analytical Pathology, Helmholtz Centre Munich, German Research Center for Environmental Health, Neuherberg, Germany. [13]Institute for Diabetes and Obesity, Helmholtz Diabetes Center, Helmholtz Centre Munich, German Research Center for Environmental Health, Neuherberg, Germany. [14]Department of Clinical Nutrition, German Institute of Human Nutrition Potsdam-Rehbruecke, Nuthetal, Germany. [15]Department of Endocrinology, Diabetes and Nutrition, Campus Benjamin Franklin, Charité University of Medicine, Berlin, Germany. [16]Biomineral Research Group, Department of Veterinary Medicine, University of Cambridge, Cambridge, UK. [17]Research Group Molecular Nutritional Medicine, Department of Molecular Toxicology, German Institute of Human Nutrition Potsdam-Rehbruecke, Nuthetal, Germany. [18]Department of Medicine, University of Leipzig, Leipzig, Germany. ✉email: anastasia.georgiadi@helmholtz-muenchen.de; stephan.herzig@helmholtz-muenchen.de

