## [Peer Review File · Nature Communications]

REVIEWERS' COMMENTS

Reviewer #1 (Remarks to the Author):

NCOMMS-20-45399-T

I carefully read both the reviewers' comments and the replies of the authors. The authors made an impressive effort to address all the suggestions and comments made by the reviewer using an impressive range of methods. Overall, this is a very interesting manuscript. There are only a few - very minor - points that should be addressed before publication.

Minor points

- Line 81 abstract: "orphan adhesion GPR116" please rephrase "orphan adhesion GPCR GPR116"
- line 153: please also give the % change for the 6 week HFD sFNDC4 levels in the text.
- line 156: isn't there a negative correlation between sFNDC4 serum levels and glucose levels as well as "glucose tolerance"? please clarify in the text "association" might be a bit misleading.
- fig. 2e: the levels of Fndc4 in liver injected with the control AAV are much lower than those depicted in fig. 1a; is the difference between "trunk blood as opposed to tail blood" the explanation? The authors might want to rephrase, because in its present form it might not be clear to the quick reader.

Reviewer #2 (Remarks to the Author):

The authors have adequately responded to my concerns.

Rebuttal

REVIEWERS' COMMENTS

Reviewer #1 (Remarks to the Author):

NCOMMS-20-45399-T

I carefully read both the reviewers' comments and the replies of the authors. The authors made an impressive effort to address all the suggestions and comments made by the reviewer using an impressive range of methods. Overall, this is a very interesting manuscript. There are only a few - very minor - points that should be addressed before publication.

Minor points

-Line 81 abstract: "orphan adhesion GPR116" please rephrase

-line 153: please also give the % change for the 6 week HFD sFNDC4 levels in the text.

-line 156: isn't there a negative correlation between sFNDC4 serum levels and glucose levels as well as "glucose tolerance"? please clarify in the text "association" might be a bit misleading.

-fig. 2e: the levels of *Fndc4* in liver injected with the control AAV are much lower than those depicted in fig. 1a; is the difference between "trunk blood as opposed to tail blood" the explanation? The authors might want to rephrase, because in its present form it might not be clear to the quick reader.

Answer from Authors : We thank the reviewer for the positive comments and for the suggested clarifications. Below we respond point- to-point to the Minor points of the reviewer :

- **Line 81: At the Abstract we have replace the "orphan adhesion GPR116" with "orphan adhesion GPCR GPR116". Now this is in line 67 and line 86.**
- **Line 153 – we write now in the text the exact % change of sFNDC4 levels after 6weeks HFD (this is now in new line 159). This is 10% after 1 week and 6% change after 6 weeks.**
- **Line 156: Indeed the reviewer is correct on the fact that there is a negative correlation between the levels of FNDC4 and glucose levels. However lower glucose levels under the tests presented in Figure 1b,c indicate higher or better glucose tolerance.**
Now we write in new line 147: ' Liver *Fndc4* mRNA levels showed an inverse correlation with fasting blood glucose levels (Fig.1b and Supplementary Table1) and blood glucose levels after a 2 h oral glucose tolerance test (OGTT) (Fig.1c and Supplementary Table1) in lean healthy individuals. Since lower levels of blood glucose in those tests indicate increased glucose tolerance, these findings

proposed a positive correlation between FNDC4 levels and glucose tolerance, as well as insulin sensitivity.

In addition we have replaced the word 'association' with 'correlation' as it is shown directly in the figure. The latest is found in new line 162.

- In fig.2 e we show mRNA levels of Fndc4, as well as in the fig. 2b data are $2^{\Delta\Delta Ct}$ values expressed relative to the Control group for each group. Therefore, in the previous version of our manuscript the Control group for each organ (ChowAAVshcontrol for Fig 2b and Fig.2e is set at 1) and comparing the mRNA levels of Fndc4 in between organs as it is in Fig 1a was not possible. To follow the reviewer's recommendation in the revised version of our manuscript we now present in Fig. 2b and 2e , $2^{\Delta\Delta Ct}$ values for all organs relative to the control group (ChowAAVshControl) of sk.muscle, as it was also presented in Fig. 1a. As a result of it now in Fig.1a and Fig2b, e one can compare the mRNA of Fndc4 in between organs and immediately see that the liver levels of Fndc4 mRNA are much higher compared to gWAT and sk. muscle in all 3 figures. Some difference in absolute differences indeed is present between Fig.1a and Fig. 2a, e which has to do with the levels of the housekeeping gene in all 3 tissues which are not exactly the same in all figures. We tried to normalize with housekeeping genes that were very similar in all tissues, but this is a common problem when comparing gene expression between different tissues. In any case we provide in the Source Data raw Ct values, so one can see indeed that the differences we report are true, independently of normalization to any housekeeping gene. Additionally some other parameters such as the age of the mice shown in the two different figures and the AAV treatment could possibly account for variations in the absolute differences between organs in Fig. 1a, 2b,e.
The comment of the reviewer on "trunk blood as opposed to tail blood" does not apply here, since this trunk tail differences are referring to circulating levels of sFNDC4 protein in our manuscript and not mRNA levels in tissue.

Reviewer #2 (Remarks to the Author):

The authors have adequately responded to my concerns.

Answer from Authors

We thank the reviewer for supporting our manuscript and we have nothing to add.